# SepN is a septal junction component required for gated cell–cell communication in the filamentous cyanobacterium *Nostoc*

Ann-Katrin Kieninger[1,3], Piotr Tokarz[2,3], Ana Janović [1], Martin Pilhofer[2], Gregor L. Weiss [2] ✉ & Iris Maldener [1] ✉

Multicellular organisms require controlled intercellular communication for their survival. Strains of the filamentous cyanobacterium *Nostoc* regulate cell–cell communication between sister cells via a conformational change in septal junctions. These multi-protein cell junctions consist of a septum spanning tube with a membrane-embedded plug at both ends, and a cap covering the plug on the cytoplasmic side. The identities of septal junction components are unknown, with exception of the protein FraD. Here, we identify and characterize a FraD-interacting protein, SepN, as the second component of septal junctions in *Nostoc*. We use cryo-electron tomography of cryo-focused ion beam-thinned cyanobacterial filaments to show that septal junctions in a *sepN* mutant lack a plug module and display an aberrant cap. The *sepN* mutant exhibits highly reduced cell–cell communication rates, as shown by fluorescence recovery after photobleaching experiments. Furthermore, the mutant is unable to gate molecule exchange through septal junctions and displays reduced filament survival after stress. Our data demonstrate the importance of controlling molecular diffusion between cells to ensure the survival of a multicellular organism.

Multicellular organisms rely on cell–cell communication for proper growth and survival. Plants and metazoans use plasmodesmata and gap junctions, respectively, to exchange molecules between sister cells. A multicellular lifestyle in bacteria evolved in filamentous cyanobacteria, especially in those, which perform cellular differentiation under specific environmental conditions. Within this group, *Nostoc* sp. PCC 7120 (also known as *Anabaena* sp. PCC 7120; hereafter *Nostoc*) is a model organism to study bacterial cell–cell communication[1]. As a Gram-negative cyanobacterium, *Nostoc* has an outer membrane, which encompasses the entire filament without entering the septum after cell division[2]. The peptidoglycan (PG) represents a contiguous giant macro molecule, which surrounds every single cell, but is fused inside the septum between the cells[3,4]. Molecules determined to traffic

intercellularly along the filament have to cross the cytoplasmic membrane of each cell and the septal PG[1,5,6].

To enable this molecular exchange, *Nostoc punctiforme* and *Nostoc* exhibit semi-regular 20 nm wide perforations in their septal PG disk, which is referred to as the nanopore array[3,7]. Cell wall amidases are responsible for drilling these 50–150 nanopores per septal PG disk, which are essential for intercellular communication[8–10] (reviewed in:[11]). Previously described cell–cell connections[12,13] likely traverse these nanopores and directly join the cytoplasm of neighboring cells[5,14]. These structures are most probably of proteinaceous nature[15,16] and were named septal junctions (SJs)[5,14]. Recently, we revealed the in situ architecture of SJs in *Nostoc* by imaging focused ion beam (FIB)-thinned filaments with cryo-electron tomography (cryoET)[17]. The SJs

[1]Interfaculty Institute of Microbiology and Infection Medicine Tübingen, Organismic Interactions, University of Tübingen, Auf der Morgenstelle 28, 72076 Tübingen, Germany. [2]Department of Biology, Institute of Molecular Biology & Biophysics, Eidgenössische Technische Hochschule Zürich, Otto-Stern-Weg 5, 8093 Zürich, Switzerland. [3]These authors contributed equally: Ann-Katrin Kieninger, Piotr Tokarz. ✉e-mail: gregor.weiss@mol.biol.ethz.ch; iris.maldener@uni-tuebingen.de

exhibit a five-fold symmetric cytoplasmic cap module, a cytoplasmic membrane-embedded plug module and a tube flexible in length that spans the septal PG connecting to the cap/plug module of the opposite cell[17]. The septal protein FraD[18] localizes near the plug module and likely represents a structural SJ component. This became evident, since plug and cap modules are absent in a ΔfraD mutant and an additional density is detectable near the plug when a green fluorescent protein (GFP) is fused to FraD[17]. Mutants in other known septal proteins, AmiC1, FraC, SepJ, SjcF1, SepI and GlsC have a similar phenotype of impaired nanopore array, reduced rate of molecular exchange, filament fragmentation (except for AmiC1 and SjcF1) and inability of diazotrophic growth[9,18–24]. However, mutants lacking SepJ, SjcF1, AmiC1, or FraC show no altered SJ architecture, and therefore are most likely no structural SJ components[17].

Under standard culturing conditions, the molecular exchange along the filament is driven by simple diffusion and can be traced using fluorescent dyes in fluorescence recovery after photobleaching (FRAP) experiments[25,26]. However, interconnection of all cells within a filament also constitutes a risk for the organism when individual cells are attacked by predators[27], burst due to shear forces[28], undergo a programmed cell death in response to environmental stresses[29,30] or die of senescence. In such cases, isolation of the injured cell by abolishment of molecular exchange seems to be essential for ensuring the survival of the remaining filament. Hence, communication is indeed interrupted between senescent terminally differentiated heterocysts and vegetative cells[7]. Furthermore, the SJ cap module undergoes a conformational change when facing stress conditions like disruption of the proton motive force, oxidative stress, or prolonged darkness. This leads to a closed cap structure, which is accompanied by abolished intercellular exchange[17]. Importantly, the cap can switch back to the open, communication-allowing conformation when conditions are more favorable. This renders SJs to gated cell–cell connections, functionally analogous to metazoan gap junctions[17]. The importance of the cap and plug module on SJ gating is further demonstrated by a ΔfraD mutant, which lacks the cap and plug modules and is unable to abolish communication when exposed to stress conditions[17].

Although a variety of proteins involved in cell–cell communication were identified, their interplay in the process of assembling the communication apparatus is still poorly understood and the structural SJ components remain, with one exception, unknown[1,31,32]. Here, we identified a second structural component of septal junctions and demonstrated its essential role in gating intercellular communication.

## Results and discussion

### Identification of SepN as a putative interaction partner of FraD
In a previous study, we show that FraD is a potential structural component of septal junctions[17]. In order to identify further proteins of the multimeric SJ protein complexes, we performed co-immunoprecipitation experiments (co-IPs) using FraD or GFP-FraD as bait. All independently performed co-IPs (in total 6) are summarized in Supplementary Table 1. Detected proteins were considered as putatively interacting with FraD, when they were 100 times more abundant in the sample than in the control and if they were conserved only in filamentous cyanobacteria similarly to FraD (full list of hits in Supplementary Data 1). The most promising candidate was the protein encoded by gene all4109, since it was detected in all crosslinked and non-crosslinked samples in every co-IP experiment with high abundance. Furthermore, the protein was described as a signature protein for the order Nostocales[33]. We renamed All4109 to SepN and will use the latter name hereafter. As a control, we performed co-IPs with a SepN derivate as bait (see Methods) resulting in FraD as the most abundant protein hit (Supplementary Data 1).

The sepN gene is the second in a predicted operon together with gene all4110 (MicrobesOnline Operon Prediction) (Supplementary Fig. 1). The gene product of all4110 is annotated as magnesium

transport protein CorA that mediates the influx of magnesium ions, whereas sepN encodes an unknown protein of 235 aa. However, a putative promoter sequence inside the upstream gene all4110 was observed previously and could allow individual transcription of sepN[34]. The SepN protein is predicted to contain one transmembrane domain (aa 7–29, InterPro[35]), with the N-terminal end facing the cytoplasm and the C-terminal part ranging into the periplasm (Protter[36]). Pfam domain analysis[37] revealed low similarity of the N-terminal part to an Apo-citrate lyase phosphoribosyl-dephospho-CoA transferase (E-value 0.0049). By querying the protein interaction database STRING, we found that with one exception, SepN always co-occurs with FraD. Conserved homologs of SepN can be found in filamentous cyanobacteria of the families Oscillatoriaphycideae, Nostocaceae, and Stigonematales. The predicted absence of enzymatic domains and the FraD-like taxonomic distribution in filamentous cyanobacteria is in line with a presumable role of SepN in cell–cell communication.

### SepN localizes to the septum
Proteins related to the cell–cell communication system of Nostoc, like FraC, FraD, SepJ, AmiC1 and SjcF1, localize to the septum between neighboring cells[18–21,24]. To visualize the subcellular localization of SepN, a single recombinant C-terminal fusion to superfolder (sf) version of GFP was inserted into the genome replacing the wild type (WT) gene. In this engineered strain SR834, the GFP-fusion protein was expressed from its native promoter avoiding overexpression artifacts.

Analysis of the sepN-sfgfp strain via fluorescence light microscopy (fLM) revealed focused fluorescence foci in mature septa of adjacent vegetative cells similarly to foci observed for GFP-tagged FraD [GFPmut2-FraD (CSVT2.779)] (Fig. 1a). As fluorescence background control, WT cells not expressing any GFP were imaged with identical settings. No GFP foci could be observed in constricting septa of dividing cells of the sepN-sfgfp strain (Fig. 1b). This observation is similar to what was observed for FraD[18] and can be explained by the fact that the nanopore array is only established once the septum is fully closed. Interestingly, other SJ-related proteins like AmiC[24], SepJ[19], FraC[18], and SepI[22] migrate with the FtsZ ring during septum constriction and were suggested to play a role in the maturation of the nanopore array and not constituting a structural component of SJs[1].

To determine if SepN also localizes to septa between vegetative cells and heterocysts, a 4 days nitrogen stepdown of a sepN-sfgfp culture was performed. We detected constricted fluorescent foci in heterocyst-vegetative cell septa of mature terminal and intercalary heterocysts (Fig. 1c). A dispersed signal was observed surrounding the polar plugs (also known as nodules), which are intracellular assemblies of the nitrogen storage molecule cyanophycin at the poles of heterocysts[38]. A similar re-localization of septal proteins during heterocyst formation was also observed for SepJ[39] and could be a result of the constricting septum.

### SepN requires FraD for septal localization
As we identified SepN as potential interaction partner of FraD, we wanted to obtain further insights whether the septal localization of FraD was influenced by the presence of SepN or vice versa. For this, we first created a sepN deficient mutant (sepN⁻ mutant; DR825) by inserting the neomycin resistance cassette C.K3t4 into the sepN gene. Next, we visualized the septal localization of FraD in the WT and in the sepN⁻ mutant background via immunolocalization. While the overall localization of fluorescent foci to the septum was similar in both strains, some septa of the sepN⁻ mutant showed somewhat broader septal foci compared to the WT (Fig. 2a). Now, we aimed to localize SepN in a ΔfraD mutant. Interestingly, the focused fluorescent foci in the septum observed for SepN-sfGFP in the WT background were absent in the majority of ΔfraD cells (Fig. 2b, c). Consequently, localization of SepN might be dependent on the presence of FraD, but not vice versa. An alternative explanation might be that the SepN-sfGFP

fluorescence signal was below the detection limit in the Δ*fraD* mutant, since this mutant has a reduced number of septal junctions[7,17].

## A *sepN⁻* mutant can grow diazotrophically but has impaired cell−cell communication and gating of septal junctions

Since mutants in SJ-related proteins were impaired in diazotrophic growth, we investigated the growth behavior of the *sepN⁻* mutant under nitrogen limiting conditions. Growth on nitrate as nitrogen source was similar compared to WT, however, when nitrate was depleted, the mutant showed slower growth on solid and liquid media (Fig. 3a, b). When analyzing the mutant heterocysts with light and electron microscopy, they did not show aberrant morphology, forming the typical heterocyst envelope and restricted septum (Fig. 3c; Supplementary Fig. 2). The filament length was slightly reduced during diazotrophic growth compared to the WT (Supplementary Fig. 3), but was clearly different from the Δ*fraD* mutant, which heavily fragments under these conditions[18]. Even though SepN-sfGFP localizes to heterocyst-vegetative septa in the WT (Fig. 1c), it does not seem to be essential for heterocyst formation and function, in contrast to FraD.

The Δ*fraD* mutant and several other mutants associated with the cell−cell communication apparatus exhibit a lower rate of molecular exchange compared to the WT[1]. To obtain first insight on a potential

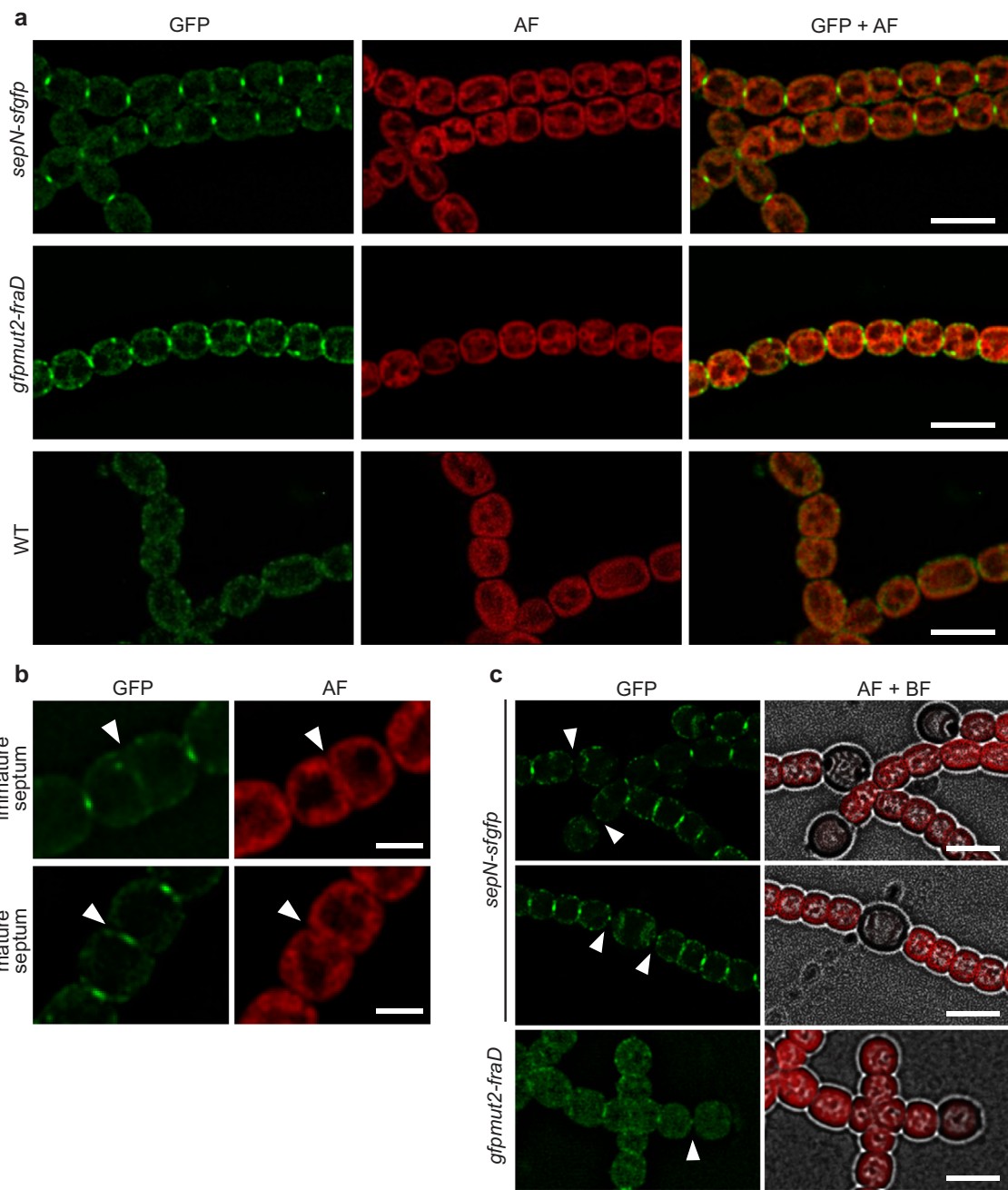

**Fig. 1 | SepN localizes to the septum. a** Green fluorescent foci of SepN-sfGFP were detected in the septum between vegetative cells. The localization of fluorescent foci is similar to GFPmut2-FraD. WT cells were included as a negative control. Shown are fluorescence micrographs of GFP, autofluorescence (AF) coming from the photosynthetic pigments, and an overlay of both. Bars, 5 μm. **b** No fluorescence could be detected for SepN-sfGFP in immature septa of dividing cells but appeared early after cell division. Arrowheads point to the septum of a dividing cell. Bars, 2 μm. **c** SepN-sfGFP was detected in the septum of terminal (upper panel) and intercalary (middle panel) heterocysts. The *gfpmut2-fraD* strain was included for comparison. Cultures were grown for 4 days on nitrogen-depleted agar plates before imaging. Arrowheads point to the heterocyst-vegetative cell septa. AF autofluorescence, BF bright field. Bars, 5 μm.

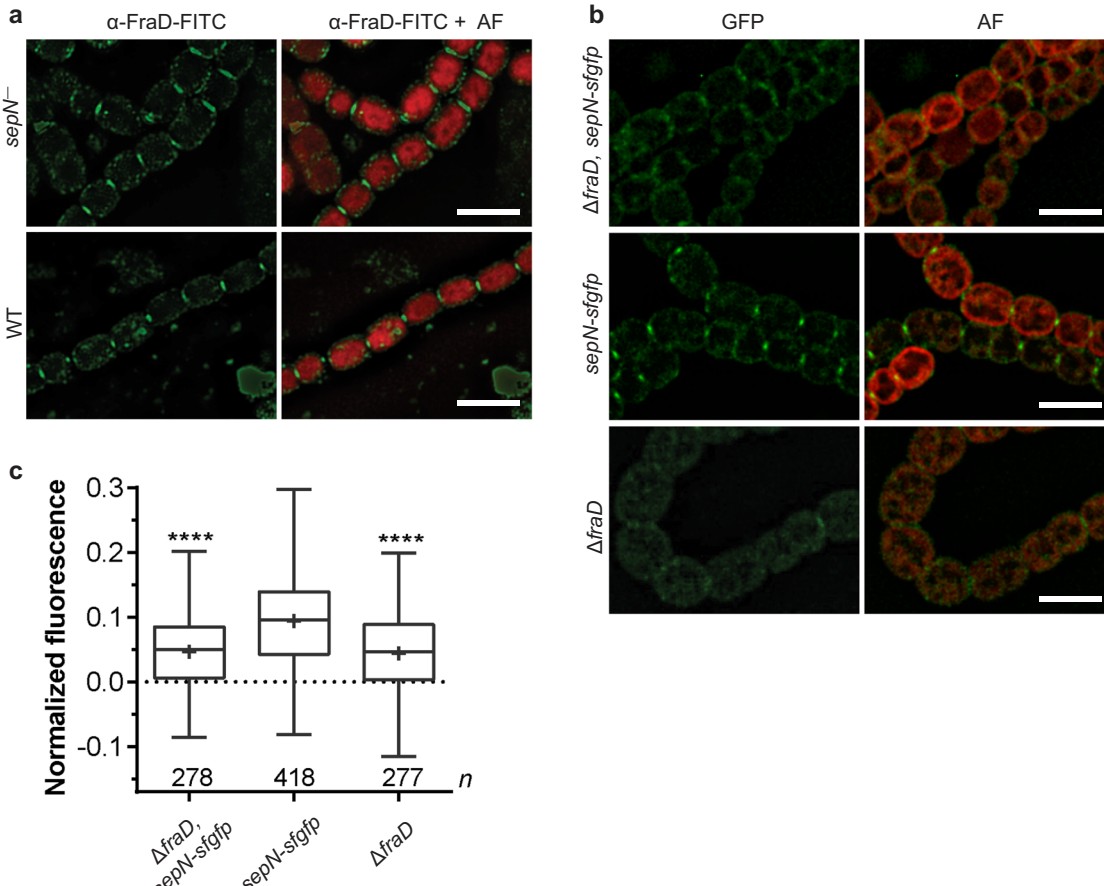

**Fig. 2 | SepN requires FraD for septal localization. a** Immunolocalization of FraD in the *sepN*⁻ mutant and in the WT revealed septal localization of FraD in both strains. The foci in the *sepN*⁻ mutant, however, seemed to be less constricted compared to the WT. A primary α-FraD and FITC-coupled secondary antibodies were used for localizing FraD. Shown are representative 3D-deconvoluted fluorescence images. Bars, 5 μm. FITC, fluorescein isothiocyanate; AF, autofluorescence. **b** Fluorescence light microscopy images of SepN-sfGFP in the WT and in the Δ*fraD* mutant showed septal localization of SepN only in the WT. This indicates that septal localization of SepN might be dependent on FraD. Plasmid pIM834 encoding for *sepN-sfgfp* was single recombinantly inserted into the genomic *sepN* gene of the Δ*fraD* mutant or WT background. Representative 3D-deconvoluted micrographs are shown. The contrast was enhanced for better visualization. Bars, 5 μm. **c** The mean fluorescence in the septa of the GFP-fusion strains shown in (**b**) was quantified as described in the methods. Septal fluorescence was significantly reduced in Δ*fraD* (control) and Δ*fraD sepN-sfgfp* mutant strains compared to the *sepN-sfgfp* strain. Box and whiskers projections show min to max values, upper and lower box border marks interquartile range, the line in the box marks the median. The numbers of measured septa (*n*) are indicated below the boxes. Statistical significance was tested via one-way ANOVA followed by Dunnett's multiple comparison test compared to strain *sepN-sfgfp*. ****$p \leq 0.0001$. Source data are provided as a Source Data file.

---

involvement of SepN in intercellular communication, we traced the cell-to-cell diffusion of the fluorescent dye calcein by FRAP measurements in the *sepN*⁻ and *sepN-sfgfp* mutants. The FRAP measurements showed that the fluorescence recovery rate constant *R* in the *sepN*⁻ mutant was significantly reduced, comparable to the Δ*fraD* mutant (Fig. 3d). Complementation of the *sepN*⁻ mutant with SepN expressed from a self-replicating plasmid under the control of its native promoter (strain DR825.848) restored a WT-like communication rate. Interestingly, the *sepN-sfgfp* strain showed a similar *R* as the knockout mutant. Slower dye exchange of the GFP-fusion strain might indicate steric hindrance of diffusion of molecules through SJs by GFP, similar to the recently described GFP-FraD mutant[17].

As cell–cell communication was slower in the *sepN*⁻ mutant, we wondered if SepN was also essential for gating intercellular molecular exchange. SJ closure can be triggered by disruption of the proton motive force[17] and therefore we treated cells of the *sepN*⁻ mutant, the complemented mutant, and the SepN-GFP-fusion strain with the protonophore CCCP prior to FRAP analysis. The FRAP responses were assigned to one of four groups as described previously, which are now

communication, slow increase of fluorescence recovery, recovery only between two cells and full recovery[17]. Communication under standard growth conditions was possible in all mutants, since the majority of calcein-stained cells showed a full recovery response after bleaching (Fig. 3e). Treatment of the *sepN*⁻ mutant with 50 μM CCCP led to a no communication response in only 16% of the analyzed cells, which was in contrast to 85% of non-communicating cells in the WT. A full-recovery phenotype in 63% of the CCCP-treated cells of the *sepN*⁻ mutant was very similar to the Δ*fraD* strain, which showed a fraction of 61% communicating cells after CCCP treatment. The inability to regulate intercellular communication in the Δ*fraD* mutant was linked to the absence of the SJ cap and plug module[17]. Accordingly, the strongly affected SJ gating in the absence of SepN suggested a role of this protein as an assembly factor or structural element of the SJ complex. Since SJ closure was restored in the complemented mutant, the phenotype was indeed caused by lack of the SepN protein. SJs in the complemented mutant also reopened after removal of CCCP, yet less efficient as in the WT. The *sepN-sfgfp* mutant was able to gate cell–cell communication, however, the fraction of non-communicating cells

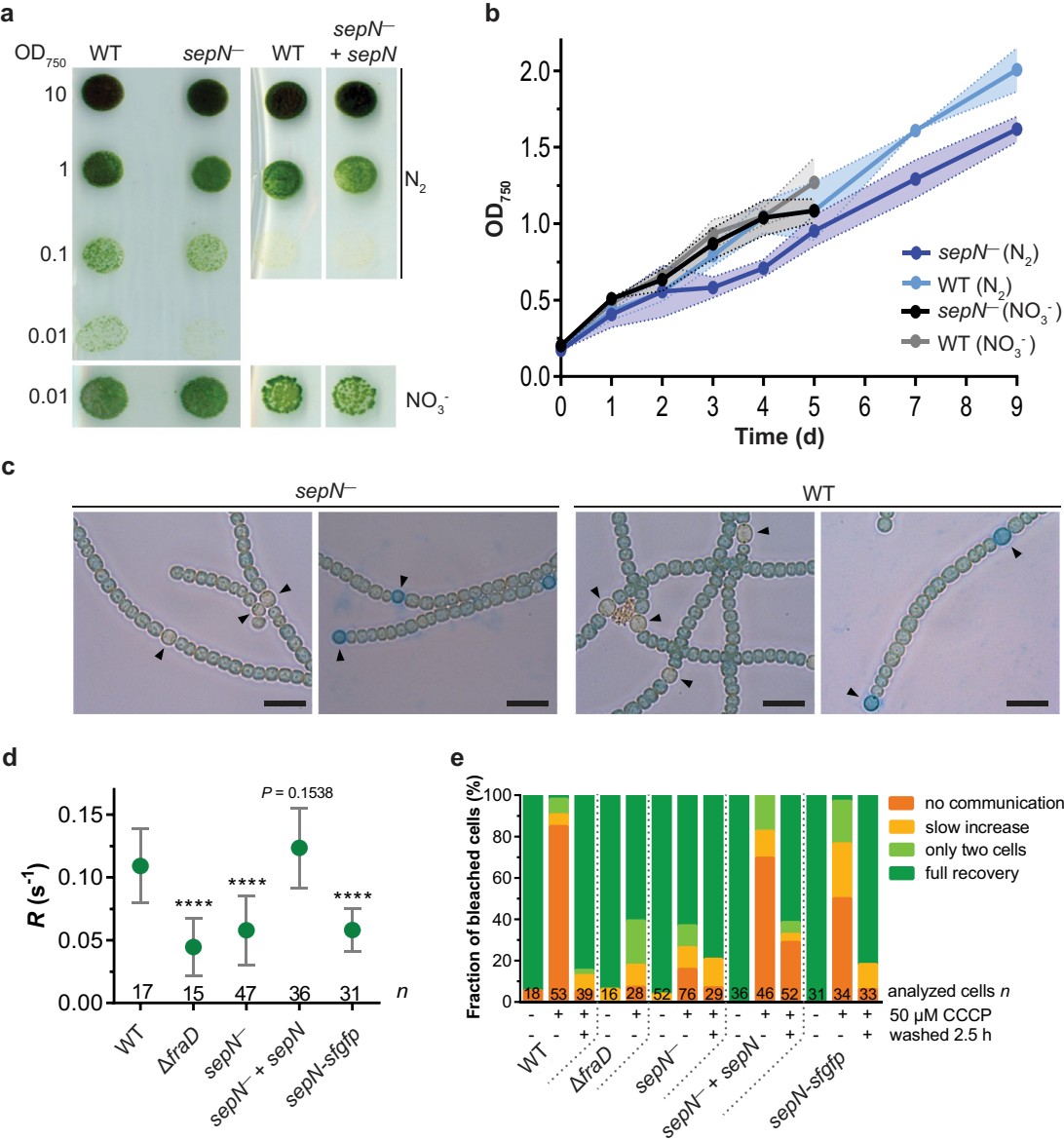

**Fig. 3 | A *sepN⁻* mutant can grow diazotrophically but has impaired cell–cell communication and gating of septal junctions. a** Spotted colonies of the *Nostoc* WT, the *sepN⁻* mutant and the complemented mutant (*sepN⁻ + sepN*) on nitrate-free medium (N₂) showed diazotrophic growth of *sepN⁻* mutant. Images were taken after 7 days under standard cultivation conditions. **b** The growth rate in diazotrophic conditions is decelerated in a *sepN⁻* mutant. No differences could be observed in the growth of the mutant and the WT in the presence of nitrate (experiment was repeated twice with biologically replicates for WT and four times for *sepN⁻* mutant). Mean values of OD₇₅₀ measurements (connected by solid line) +/− standard deviation (shaded area) are shown. Two-sided *t*-test was performed (*p*-value = 0.8446 or 0.5322 for nitrate and nitrate-free conditions, respectively). Source data are provided as a Source Data file. **c** Liquid cultures of WT and *sepN⁻* mutant were analyzed via light microscopy 48 h after transfer of the cultures to nitrate-free medium. The heterocyst polysaccharide layer was stained with Alican blue (respective right images). Both strains were able to form heterocysts (indicated by black arrowheads). Bars, 10 µm. Experiment was performed twice with similar results. **d** The fluorescence recovery rate constant *R* of diverse *Nostoc* strains after FRAP treatment. The *sepN⁻* mutant as well as the sepN-sfGFP strain

revealed a significantly reduced fluorescence recovery rate constant *R* compared to WT and similar to the Δ*fraD* mutant. *R* was calculated from untreated cells showing full recovery in FRAP experiments. Data for the Δ*fraD* strain were taken from[17]. Mean values (green dot) +/− standard deviation (error bars) are shown. The numbers below indicate the number of analyzed cells (*n*) from different filaments. Statistical significance was tested via one-way ANOVA followed by Dunnett's multiple comparison test compared to WT without adjustments. ****$p \leq 0.0001$. Non-significant *p*-values are indicated above. Cumulated results from at least two independent cultures are shown. Source data are provided as a Source Data file. **e** FRAP-responses of calcein-stained untreated, CCCP-treated (90 min, 50 µM), and washed cells after CCCP treatment were assigned to one of four groups indicated by the color scheme. Only 16% of the analyzed *sepN⁻* mutant cells showed a no communication response, which was in contrast to 85% of non-communicating cells in the WT. The numbers within the bars indicate the number of analyzed cells (*n*) from different filaments. Data were cumulated from five independent experiments representing biological replicates for *sepN⁻*, four for *sepN⁻ + sepN*, three for WT as well as for *sepN-sfgfp* and two for Δ*fraD*. Data for Δ*fraD* strain were taken from[17]. Source data are provided as a Source Data file.

upon CCCP treatment was reduced by 35% in comparison to the WT (Fig. 3e). The large GFP-tag might therefore interfere with other structural components of SJs or sterically affect the process of SJ closure.

**The nanopore arrays of the *sepN⁻* mutant and the WT are similar**
To further investigate a potential role of SepN during SJ assembly, we set out to analyze the nanopore array of *sepN⁻* mutant filaments. Mutants with a reduced nanopore array usually display a reduced rate

of molecular diffusion (reviewed in[1]). Since the diffusion rates in the Δ*fraD* and the *sepN⁻* mutants were severely reduced (Fig. 3d), we purified septal PG discs and compared the respective nanopore array to WT septa. Deletion of *fraD* led to a significantly diminished number of nanopores per septum (Fig. 4a, b). In contrast, disruption of *sepN* or fusion to GFP did not alter the number of nanopores compared to the WT. The diameters of PG disks and of nanopores were significantly enlarged in the Δ*fraD* mutant, whereas the *sepN* mutants revealed similar PG disks compared to the WT. The only difference was a slight increase in nanopore diameter in the *sepN⁻* mutant (Fig. 4c, d).

### *sepN⁻* mutant is lacking the plug module

Since the reduced communication rate in the *sepN⁻* mutant seemed not to be a result of fewer nanopores, we assumed that SepN might be a structural component of SJs and that the phenotype observed in the *sepN⁻* mutant was caused by alterations in SJs' architecture. As cryoET demonstrated to be a suitable tool to study structural changes of SJs in situ[17], we next plunge-froze *sepN⁻* mutant filaments and thinned them by cryo-FIB milling[40,41]. Tomograms of individual septal areas were acquired on the resulting lamellae (*n* = 31 tomograms) and revealed several septa spanning SJ-like complexes (Fig. 5a). A comparison between *sepN⁻* mutant and WT SJs showed striking differences in the cap/plug area. The most pronounced alteration was a missing density for the plug module (Supplementary Fig. 4). Interestingly, we noticed a reduced number of SJs per septum in the *sepN⁻* mutant (on average 4.9 SJs / tomogram; *n* = 18 tomograms with SJs) compared to WT (on average 19 SJs / tomogram; *n* = 22 tomograms with SJs), which was unexpected due to no obvious alterations in the nanopore arrays (Fig. 4). For a detailed analysis of the differences in SJ architecture of the *sepN⁻* mutant, we applied subtomogram averaging on 89 SJ ends (Fig. 5b, right panel; Supplementary Fig. 5/6). The shape of the cap resembled the previously described closed state of WT SJs (Fig. 5b, middle)[17], which is characterized by a narrower cap module compared to the WT open state (Fig. 5b, left) and by the disappearance of openings between the cap's arches. A difference map calculation (Fig. 5c) between SJs of the *sepN⁻* mutant and WT in a closed state further highlighted the missing plug module in the *sepN⁻* mutant and emphasized the similar cap architecture without any detectable openings (Fig. 5d). This observation, together with the inability of gating communication upon CCCP treatment (Fig. 3e), indicates that the cap alone might not be sufficient to completely seal SJs and the presence of the plug is required for complete closure. The plug might also be important for keeping the cap structure in its open conformation. The resulting closed cap conformation in the *sepN⁻* mutant and the reduced number of SJs per septum could explain the reduced fluorescent recovery rate *R* observed in our FRAP experiments due to a narrowed diffusion area (Fig. 3d).

### Fusion of maltose-binding protein to SepN disrupts cap assembly

We further set out to investigate if SepN was indeed a structural component of the SJ's plug module. Due to the absence of established localization tags for cryoET, other studies already successfully visualized GFP as additional densities in subtomogram averages[17,42–44]. However, when we imaged the *sepN-sfgfp* mutant strain with cryoET, the SJs resembled WT SJs in individual tomograms as well as in a subtomogram average (Supplementary Fig. 7). The absence of an additional density originating from GFP can be explained by a potential flexibility of the sfGFP-tag in relation to the SJ modules. To further increase the size of the tag to SepN, we added one copy of maltose-binding protein (MBP) to SepN-sfGFP (Supplementary Fig. 1). By using fLM imaging, we first validated that the MBP-sfGFP-fusion to SepN does not affect its septal localization (Fig. 6a). Subsequently, we acquired cryo-tomograms of the *sepN-mbp-sfgfp* mutant, which revealed aberrant SJ architecture (Fig. 6b). While the cap module was

always missing (*n* = 33 tomograms, 369 SJ ends), a plug module was still visible and strikingly, we could observe additional densities directly adjacent to the plug and emerging to the cytoplasm (Fig. 6c). A subtomogram average of these SJs ends further demonstrated that the additional densities were connected to the plug module (Fig. 6d). It is likely that the observed densities represent the MBP-sfGFP tag, therefore indicating a structural role of SepN in the plug module. The large size of the potential MBP-tag might also explain the absence of a cap module due to sterical hindrance. However, we cannot rule out a regulatory role of SepN in the correct assembly of the cap and plug module, which might be impaired due to the MBP-tag. FRAP analysis of the *sepN-mbp-sfgfp* mutant revealed that the fluorescent recovery rate constant *R* was significantly reduced compared to WT and similar to the *sepN⁻* mutant (Fig. 6e). This could be explained by the mis-arranged plug module or by a diminished diffusion area introduced by the huge tag. FRAP analysis after CCCP treatment further showed that the *sepN-mbp-sfgfp* mutant was impaired in gating cell–cell communication (Fig. 6f).

### Controlled cell–cell communication is crucial for filament survival after stress

As we now identified a potential second structural component of SJs, we were wondering how alterations in SJ's architecture affect the ability of *Nostoc* filaments to survive environmental stress. It has been previously shown that individual cells in a filament lyse after treatment with ultraviolet (UV) light, leading to filament fragmentation[29,30]. To see if the reduced capability to gate intracellular communication in the Δ*fraD* and *sepN⁻* mutants resulted in an inability to isolate injured cells after stress, we treated *Nostoc* cultures with UV light and recorded 5 h fLM time-lapse movies. The lysis of *Nostoc* cells can be easily traced, as it is accompanied by the loss of its typical red autofluorescence. Whereas in the WT mainly individual cells along the filament died (Supplementary Movie 1), it was striking that in both, Δ*fraD* (Supplementary Movie 2) and *sepN⁻* mutants (Supplementary Movie 3), numerous adjacent cells lysed collectively in a short period of time (Fig. 7a, Supplementary Fig. 8a). To quantify this observation on the filament level, we counted cell death events in the 12 fastest lysing filaments from each strain against time (data from three independent experiments). While only 24% of analyzed WT cells lysed within 5 h, 93 and 88% of cells died in the Δ*fraD* and *sepN⁻* mutants, respectively. To quantify survival on the culture level, we measured growth of WT, Δ*fraD* and *sepN⁻* cultures after UV treatment and observed a reduced survival rate of both mutants compared to WT (Supplementary Fig. 8b). To further bolster our hypothesis that the increased survival rate in the WT is a result of the isolation of injured cells from sister cells by closing SJs, we collected FRAP data on UV-treated cultures. While 84% of WT cells showed a no communication response, only 8% of Δ*fraD* mutant cells and 18% *sepN⁻* mutant cells stopped communication (Fig. 7c). This indicates that SJs indeed close after UV exposure, potentially prevent the leakage of cytoplasmic components from intact cells and the uptake of reactive oxygen molecules, enabling the survival of the rest of the filament.

In conclusion, our integrative study identified SepN as a potential structural component of SJs. SepN presumably localizes to the plug module and our mutational analysis shed light on the so far elusive role of the plug as crucial for functional gated SJs. Not only a closed cap module, but also the presence of the plug is essential to fully inhibit cytoplasmic exchange between sister cells. However, the ability of the *sepN⁻* mutant to grow diazotrophically implies that the plug-mediated gating is not required for diazotrophic growth. While the septal localization of SepN seems to be dependent on the presence of FraD, FraD still localizes to the septum in the plug-missing *sepN⁻* mutant. This challenges the assumption of FraD being a component of the plug module and therefore we speculate that FraD might serve as a linker between the plug, cap, and cytoplasmic membrane, rather than

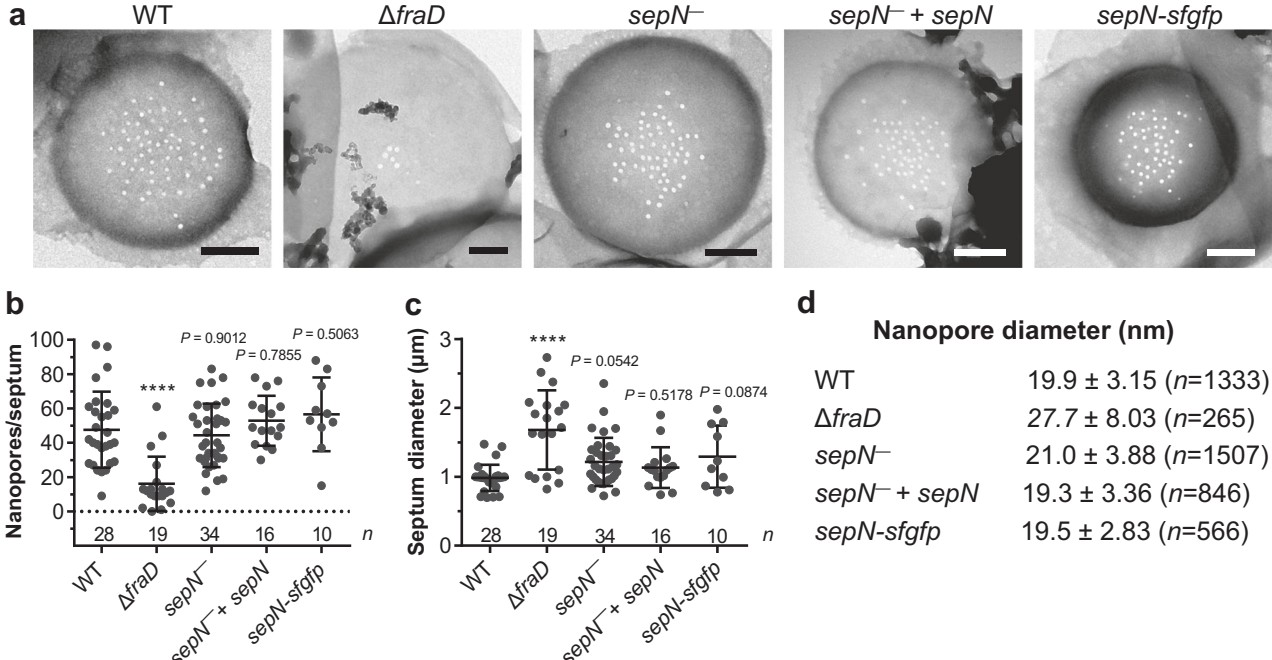

**Fig. 4 | The nanopore arrays of the *sepN⁻* mutant and WT are similar. a** Septal PG discs were isolated and analyzed via transmission electron microscopy. Shown are representative micrographs of the indicated strains. While the Δ*fraD* mutant showed an aberrant nanopore array, the nanopore arrays of *sepN* mutants were similar to WT. Bars, 250 nm. Experiment was performed four times for *sepN⁻* and twice for WT and Δ*fraD* with similar results. The experiment was performed once for *sepN⁻ + sepN* and *sepN-sfgfp*. **b, c** The diameter of nanopores (**b**) and of the septum (**c**) of mutants analyzed in **a**. The bars indicate mean and standard deviation. The numbers below indicate the number of analyzed septa (*n*). Only the Δ*fraD* mutant revealed a significantly reduced number of nanopores per septum. Each dot represents an analyzed PG septum. Statistical significance was tested via one-way ANOVA followed by Dunnett's muliple comparison test compared to WT without adjustments. ****$p \leq 0.0001$. Non-significant *p*-values are indicated above. Data were cumulated from four independent experiments representing biological replicates for *sepN⁻*, from two independent experiments for WT and Δ*fraD* and from one experiment for *sepN⁻ + sepN* and *sepN-sfgfp*. Source data are provided as a Source Data file. **d** The mean and standard deviation of the nanopore diameter cumulated from all analyzed septa are shown. The *sepN⁻* mutant showed a slight increase in nanopore diameters compared to WT, but not as severe as the Δ*fraD* mutant. The number of analyzed nanopores *n* is indicated. Data were cumulated from four independent experiments representing biological replicates for *sepN⁻*, from two independent experiments for WT and Δ*fraD* and from one experiment for *sepN⁻ + sepN* and *sepN-sfgfp*. Source data are provided as a Source Data file.

forming the plug module itself. This is in line with our structural observations, which revealed missing cap *and* plug modules in the Δ*fraD* mutant[17] and aberrant nanopore formation. The reason why FraD and not SepN is required for heterocyst function needs further investigation of the heterocyst−vegetative cell communication apparatus. Our study will therefore serve as framework to further elucidate the role of the individual SJ modules in controlling intercellular molecule exchange. The complexity of this cell−cell communication apparatus in multicellular cyanobacteria and the observation that functional SJs are important to ensure the survival of a multicellular bacterium under stress further emphasizes the striking analogy of SJs to metazoan gap junctions.

## Methods

### Strains and growth conditions

All bacterial strains used in this study are described in Supplementary Table 2.

*Escherichia coli* (*E. coli*) strains were grown in LB medium at 37 °C supplemented with 50 μg/mL kanamycin, 25 μg/mL streptomycin and 100 μg/mL spectinomycin when indicated.

Cyanobacterial strains were cultivated in liquid BG11 medium[45] at 28 °C with constant illumination at 25−40 μE m⁻² s⁻¹, either with shaking at 100−120 rpm or on BG11 medium solidified with 1.5% (w/v) Bacto agar. For co-immunoprecipitation experiments liquid cultures of 700 mL were grown in bottles by bubbling with air, mixed with $CO_2$ (2% vol/vol) at 28 °C and in constant light. Mutant strains grew in presence of the respective antibiotics at the following concentrations: 50 μg/mL neomycin, 5 μg/mL streptomycin and 5 μg/mL spectinomycin.

To induce the differentiation of heterocysts by stepdown, liquid cultures of *Nostoc* strains growing in BG11 medium, were washed three times with medium void of nitrate (BG11₀) and further cultivated in this medium under standard growth conditions.

To test the diazotrophic growth of the *Nostoc* strains, the washed and nitrate-free cultures were adjusted to optical density (OD₇₅₀) of 10, and 10 μL of serial 10-fold dilutions were spotted onto BG11 or BG11₀ plates. The plates were incubated for 7 days at standard growth conditions. To estimate the growth rate in liquid medium, nitrate-grown cultures were washed with nitrate-free medium and inoculated to OD₇₅₀ of 0.15 in 20 mL medium with or without nitrate and incubated at standard growth conditions. The OD₇₅₀ was followed during several days of growth. To analyze filament length after nitrogen stepdown, filaments were carefully collected at different time points after the shift of liquid cultures in medium void of nitrate. Samples were visualized by light microscopy and a minimum of 100 filaments per replicate and time point were counted. Mean values and SD of 2 (WT) or 3 (mutant) independent replicates are shown.

### Construction of mutant strains

DNA sequencing results were obtained from GATC biotech AG (Eurofins, Germany) and compared to reference sequences derived from the KEGG database[46]. Purification of plasmid DNA and PCR fragments was performed using the ExtractMe Kit systems (Blirt, Poland). Triparental mating (conjugation) was performed to introduce plasmids into

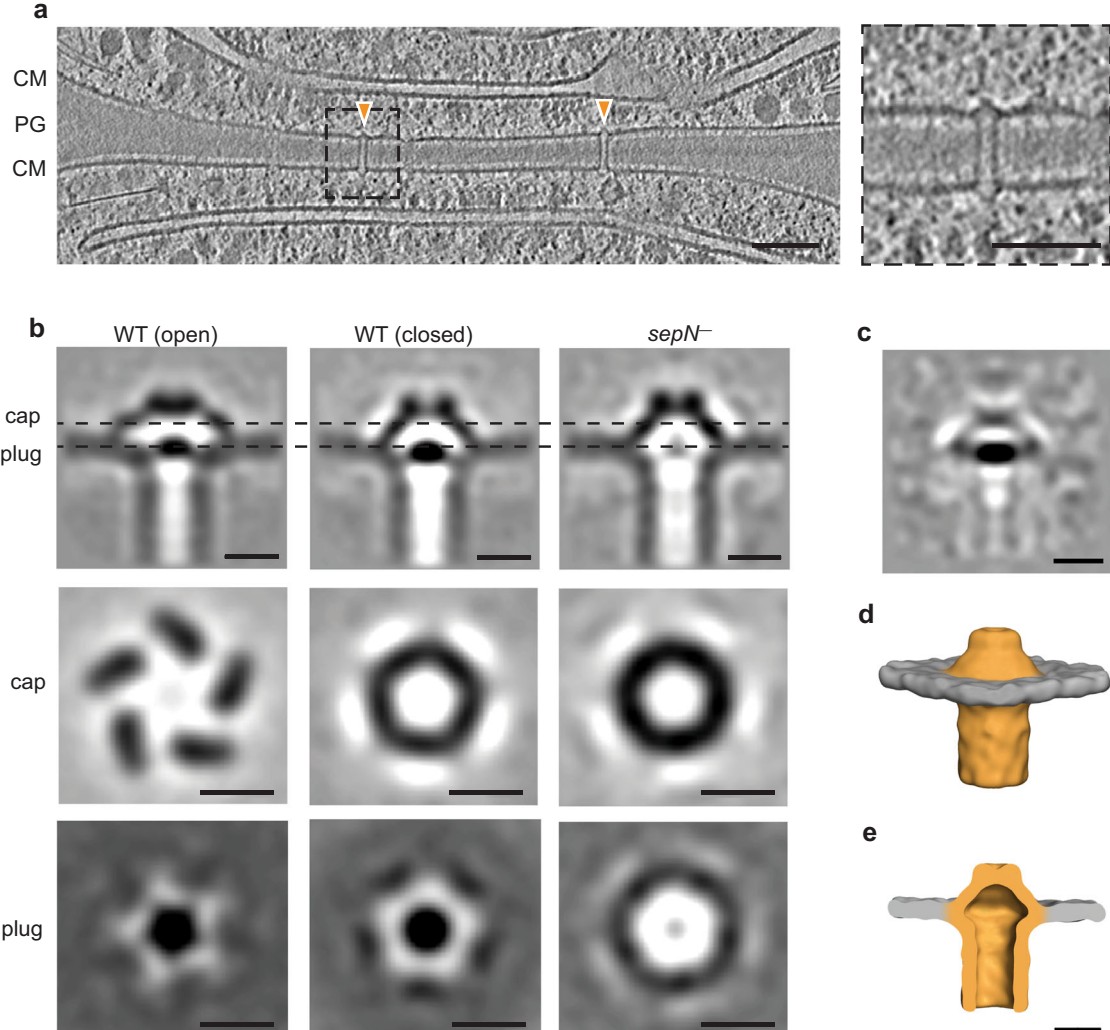

**Fig. 5 | *sepN*⁻ mutant is lacking the plug module. a** Cryo-tomogram of the septal region (shown is a 13.5 nm-thick slice) of a *sepN*⁻ mutant filament with two septum-spanning SJs (orange arrowheads). While cap and tube modules were still present, the plug module was absent in all observed SJs (*n* = 18 tomograms from 3 independent datasets of biological replicates). The dashed box indicates a magnified view on the right. CM, cytoplasmic membrane; PG, septal peptidoglycan. Bars, 50 nm. **b** Subtomogram averages of SJs from *sepN*⁻ mutant (right) and WT in open (left) and closed state (middle). In the *sepN*⁻ mutant, the plug was absent, and the cap resembled the closed state of WT SJs (middle). Shown are longitudinal and cross-sectional slices (0.68 nm) through the averages. Sliced positions (cap and plug) are indicated by dashed lines. For the initial averages, 418 (from 22

tomograms), 282 (from 18 tomograms) and 89 particles (from 18 tomograms) were picked for WT (open state), WT (closed state) and *sepN*⁻ mutant, respectively. Bars, 10 nm. **c** A difference map calculation between the subtomogram averages of SJs from CCCP-treated WT cells (closed SJ state) and *sepN*⁻ mutant cells revealed that the most prominent difference is the missing density in the plug region. Subtomogram averages were low-pass filtered to 35 Å resolution before calculating the difference map. Bar, 10 nm. **d, e** Surface representation of the subtomogram average of *sepN*⁻ mutant SJs showed no openings in the cap module (**d**), resembling the shape of WT SJs in the closed state. Sliced view of surface representation (**e**) reveals missing plug module. Bar, 10 nm.

*Nostoc* cyanobacterial cells using the *E. coli* strains J53 (RP-4)[47] and HB101 (pRL528) carrying the cargo plasmid[48]. A description of used plasmids and sequences of oligonucleotides are summarized in Supplementary Table 2 and Supplementary Table 3, respectively.

As a control for α-GFP co-IPs, the gene encoding a version of the green fluorescent protein (GFP) *gfpmut2*[49] was cloned under the control of the *fraCDE* promoter region, yielding plasmid pIM800. P*fraCDE*-*gfpmut2* was amplified with oligonucleotides 1998/2235 using pIM779[17] as template. The insert was ligated into EcoRI/BamHI digested shuttle vector pRL1049[50] and conjugated into *Nostoc*, creating strain 7120.800.

The *all4109* (*sepN*) gene was interrupted by insertion of the cassette C.K3*t4* provoking neomycin/kanamycin resistance under the control of the strong promoter P*psbA* and a transcriptional terminator. For construction of this cassette, the earlier described cassette C.K3[51]

was C-terminally fused downstream of the BamHI restriction site to the C-terminus of the bacteriophage T4 gene *32*[52], which harbors a translation tandem stop and a transcriptional terminator. The construct was ordered as a synthetic gene from Eurofins Genomics and amplified with oligonucleotides 1383/1384. To disrupt *sepN*, an upstream DNA fragment of *sepN* and a fragment within the end of the gene were amplified using oligonucleotides 2399/2397 and 2398/2400, respectively. The fragments were ligated via Gibson Assembly Cloning in a way flanking C.K3*t4* in XhoI/PstI digested pRL277 vector[53], yielding plasmid pIM825. After conjugal transfer of pIM825 into *Nostoc*, double-crossover homologous recombination with the genome and segregation of the resulting *sepN*⁻ mutant DR825 was checked via PCR using the oligonucleotides 2446/2447.

For localization studies, a C-terminal translational fusion of superfolder (sf) GFP to *sepN* was achieved by single homologous

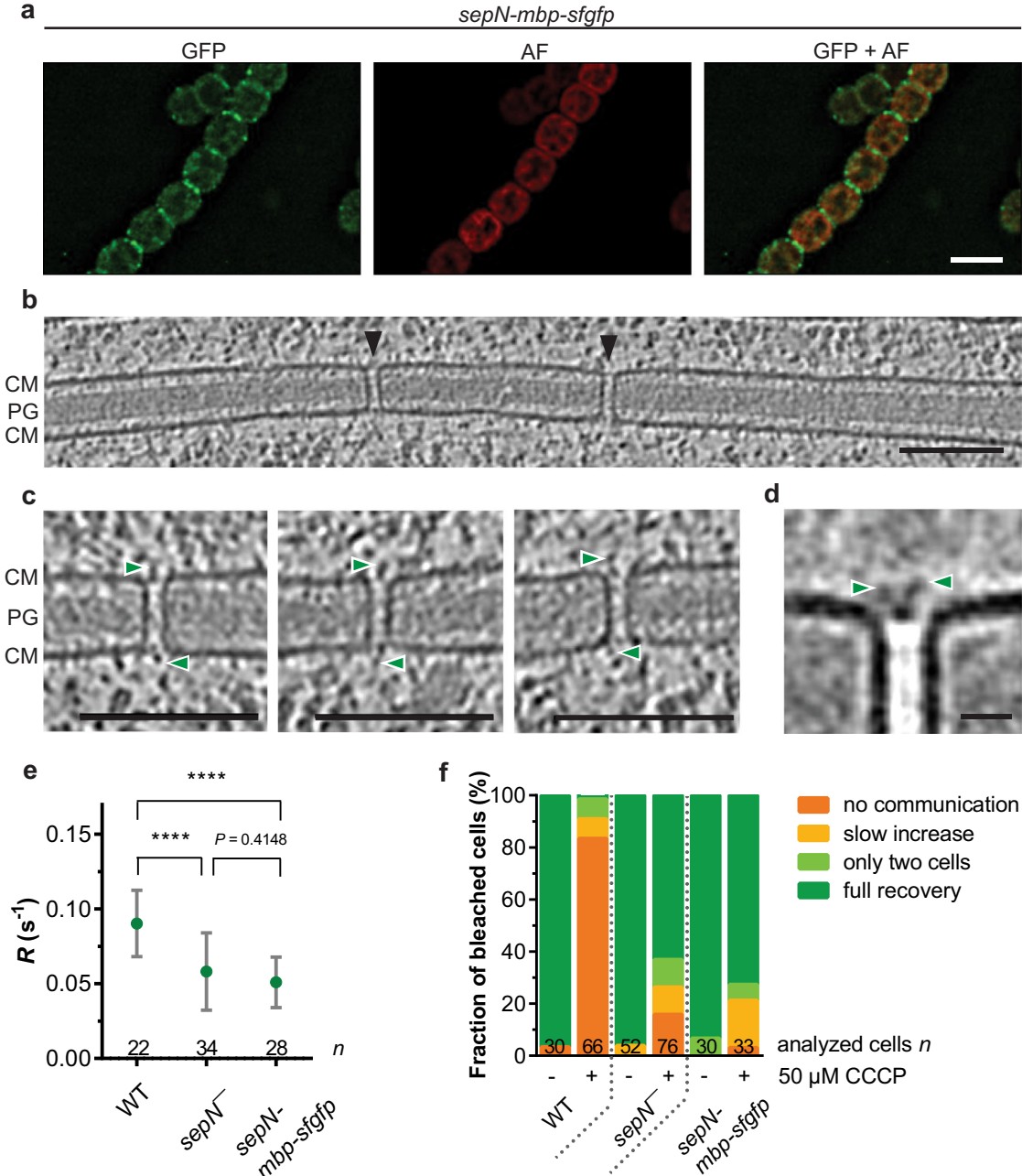

**Fig. 6 | Fusion of maltose-binding protein to SepN disrupts cap assembly and impairs gating of cell–cell communication. a** fLM analysis of a *Nostoc* filament expressing SepN-MBP-sfGFP reveals septal foci. Bar, 5 μm. GFP, gfp channel; AF, autofluorescence. Representative images of one conjugant are shown. **b** Slice through a cryo-tomogram of the septal area of *sepN-mbp-sfgfp* mutant. Arrowheads indicate SJs with altered plug structure. The cap module was always missing (*n* = 33 tomograms, 369 SJ ends from 5 independent datasets of biological replicates). Shown is a 13.5 nm-thick slice, CM, cytoplasmic membrane; PG, septal peptidoglycan. Bar, 100 nm. **c** Shown are magnified views of individual SJs of the *sepN-mbp-sfgfp* mutant in cryo-tomograms. Green arrowheads point to additional densities adjacent to the plug. Shown are 13.5 nm-thick slices. CM, cytoplasmic membrane; PG, septal peptidoglycan. Bar, 100 nm. **d** Subtomogram average of SJs from *sepN-mbp-sfgfp* mutant revealed a missing cap module and an additional density protruding from the plug towards the cytoplasm. Bar. 10 nm. **e** The fluorescence

recovery rate constant *R* of communicating cells in the *sepN-mbp-sfgfp* strain is significantly reduced compared to WT and similar to the *sepN⁻* mutant. *R* was calculated from untreated cells showing full recovery in FRAP experiments. Mean values are indicated as green dots and error bars represent +/− standard deviations. The numbers below indicate the number of analyzed cells (*n*) from different filaments. Statistical significance was tested via one-way ANOVA followed by Tukey's multiple comparison test without adjustments. ****$p \leq 0.0001$. Non-significant *p*-values are indicated above. Data were cumulated from two independent experiments with biological replicates. **f** FRAP responses of the *sepN-mbp-sfgfp* show that the mutant is unable to close SJs after CCCP treatment. Results of two independent *sepN⁻* mutant and *sepN-mbp-sfgfp* strains were cumulated. The numbers within the bars indicate the number of analyzed cells (*n*) from different filaments. Source data are provided as a Source Data file.

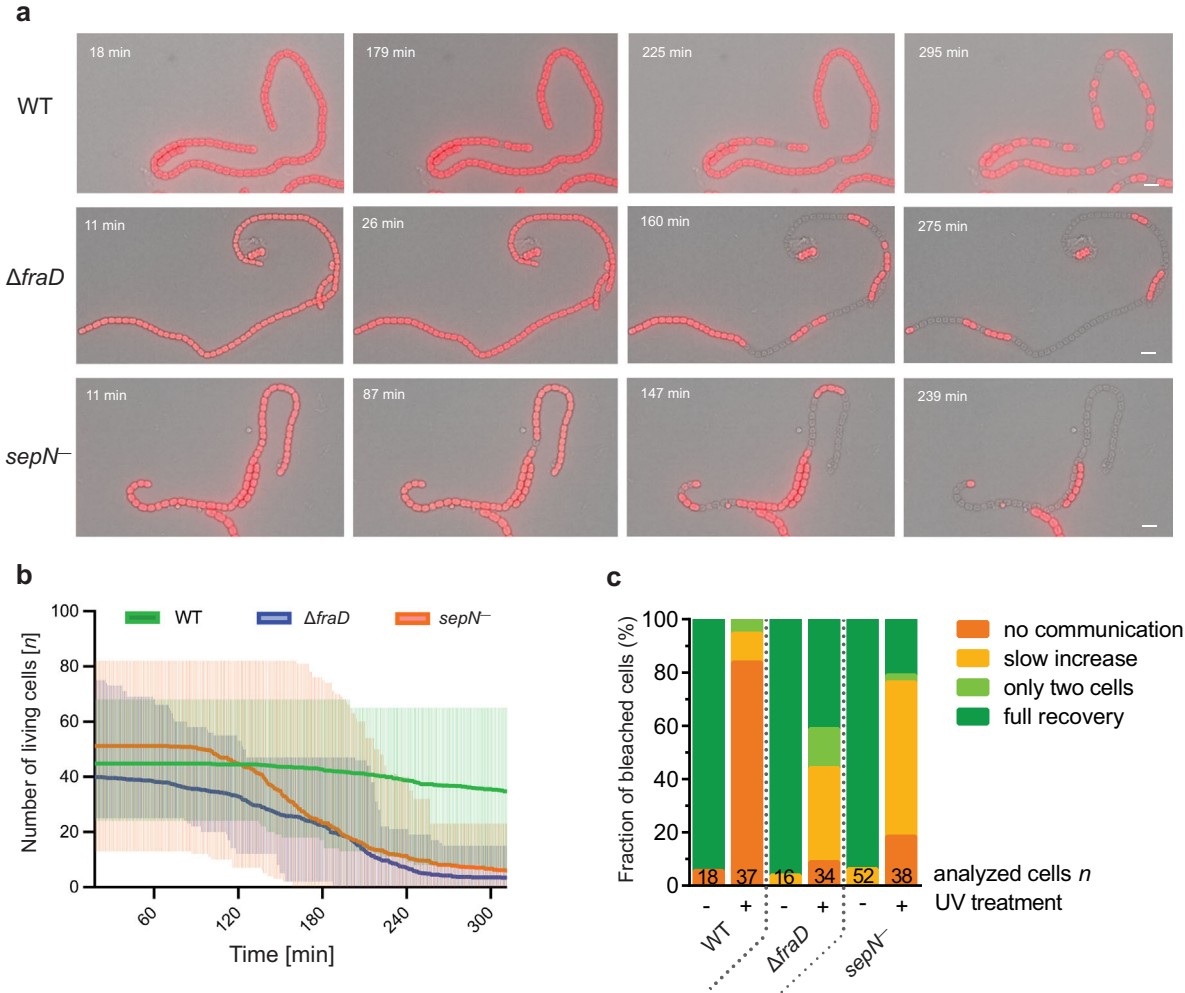

**Fig. 7 | Controlled cell–cell communication is crucial for filament survival after stress. a** Merged brightfield and fLM time-lapse of stressed *Nostoc* WT, Δ*fraD*, and *sepN*⁻ mutants after 5 min UV exposure. Living cells show red autofluorescence originating from photosynthetic pigments. With increasing time, cells within the filaments start to lyse, which results in filament fragmentation. While in the WT mainly individual cells lyse, lysis in the mutant strains occurs faster and cells often lyse in clusters of several contiguous cells. The time post-UV treatment is indicated in images. For full-length time-lapse, see Supplementary Movies 1–3. The experiment was repeated independently three times with similar results and in total 54 time-lapse movies were acquired. Bars, 10 μm. **b** Quantification of cell lysis in the twelve fastest lysing filaments from 5 h fLM time-lapse movies. In the Δ*fraD* and *sepN*⁻ mutants, lysis events increase after about 2 h post UV exposure. After 5 h, 93

and 88% of the cells in Δ*fraD* and *sepN*⁻ mutant filaments are dead, while only 24% of analyzed WT cells lysed. The solid line shows the mean number of living cells per filament. The shaded area shows the absolute number (minimum and maximum) of living cells in the analyzed filaments. Data retrieved from three independent experiments. For the lysis rate in each individual filament see Supplementary Fig. 8a. Source data are provided as a Source Data file. **c** FRAP response of WT, Δ*fraD*, and *sepN*⁻ mutants post UV treatment shows that UV stress causes SJ closure in 84% of WT cells, while in Δ*fraD* and *sepN*⁻ mutants only 8 and 18% of cells stopped communication, respectively. The numbers within the bars indicate the number of analyzed cells (*n*) from different filaments. Cumulated results from two independent UV-treated cultures are shown. Data for non-UV-treated samples were taken from Fig. 3. Source data are provided as a Source Data file.

recombination of plasmid pIM834 with the genomic DNA. The coding sequence for sfGFP was amplified using oligonucleotides 1444/1445 and the *sfgfp-gene* bearing plasmid pIM660.2 as template (Bornikoel, unpublished). The C-terminal part of *sepN* was amplified without stop codon using oligonucleotides 2450/2451. Fragments were ligated via Gibson Assembly Cloning in XhoI/PstI digested vector pRL277 and the resulting plasmid pIM834 was transferred to *Nostoc* creating mutant SR834. Mutant segregation was checked via PCR with oligonucleotides 2446/2447, what only leads to amplification in the WT, since the whole plasmid is inserted into the tested region. To localize SepN in a *fraD* mutant, single recombinant homologous recombination of pIM834 was performed in strain CSVT2[18], creating the *sepN-sfGFP* expressing *fraD* mutant CSVT2.SR834.

To complement the *sepN*⁻ mutant, the predicted promoter region of the *all4110-all4109* operon (Softberry BPROM[54]) was amplified with

oligonucleotides 2511/2512 and fused via PCR to the amplified *sepN* gene (oligonucleotides 2513/2514). The fusion product was ligated into EcoRI/BamHI digested pRL1049[50] via Gibson Assembly Cloning and the resulting plasmid pIM848 was conjugated into DR825, creating strain DR825.848.

For cryoET purposes, a bigger tag consisting of maltose-binding protein (MBP) and sfGFP was C-terminally fused to *sepN* via single homologous recombination. For this, three individual PCRs were performed to amplify *sepN* with oligonucleotides 2450/2536, 4xG-MBP using oligonucleotides 2537/2538 and 5xGS-sfGFP with oligonucleotides 1444/1445. Fragments were ligated via Gibson Assembly Cloning in XhoI/PstI digested vector pRL277. The resulting plasmid pIM855 was conjugated into *Nostoc* creating mutant SR855. Segregation of the mutant was checked via PCR as described for strain SR834.

## Co-immunoprecipitation and liquid chromatography mass spectrometry (LC-MS/MS)

Strains *Nostoc* WT, CSVT2[18], CSVT2.779[17], and 7120.800 were cultivated 7–10 days with supply of 2% $CO_2$ at 28 °C in constant light. For controls and specific conditions of independent immunoprecipitation experiments see Supplementary Table 1. Cells were washed with PBS pH 7.4, resuspended in 15/20 mL PBS to $A_{750} = 15/25$ (α-GFP-/α-FraD-pulldown) and crosslinked with 1% glutaraldehyde for 30 min at RT when indicated. In one of the α-GFP immunoprecipitations crosslinking was performed with 0.6% formaldehyde for 30 min at RT followed by two washing steps and incubation for 10 min with 1% glutaraldehyde at RT.

After washing with PBS and addition of a protease inhibitor cocktail tablet (cOmplete™, Roche), cells were lysed via three passages through a French press at 20,000 psi. Whole cells were discarded through low-speed centrifugation and membranes were collected via centrifugation at 48,000–70,000 g at 4 °C for 1 h. Membranes were solubilized in 500 μL 10 mM Tris-HCl, 100 mM NaCl pH 8 (α-GFP) or PBS pH 7.4 (α-FraD) supplemented with 1% *N*-lauroylsarcosine for 1 h at RT and peptidoglycan was digested in parallel by addition of 1 mg/mL lysozyme. Insolubilized membranes were discarded after centrifugation at 21,000 × *g* at 4 °C for 25 min. Co-IP with α-GFP magnetic beads (GFP-Trap®_MA, Chromotek) was performed following the instructions of the supplier. Solubilized membranes were incubated with the beads for 1:30 h under slow rotation at 4 °C. Bound proteins were eluted with 70 μL 2x SDS (sodium dodecyl sulfate) loading dye in Tris buffer (see above) and boiled at 95 °C for 10 min. For α-FraD immunoprecipitation, Dynabeads™ Protein G (Invitrogen) were incubated with 10 μg purified FraD antibodies in PBS for 10 min at RT. FraD antibodies were raised in rabbit against the synthetic peptide NH2-IWTGPTANPR-GYFLRKSC-CONH2 within the periplasmic part of FraD (Pineda Antikörper-Service, Berlin, Germany). The FraD antibody was validated by using the Δ*fraD* mutant CSVT2 as control. Solubilized membranes were incubated with α-FraD-bound magnetic beads for 1:30 h at 4 °C and eluted with 40 μL 50 mM glycine, pH 2.8 and 20 μL 6xSDS loading dye. Samples were boiled for 10 min at 70 °C and neutralized with 5 μL 1 M Tris, pH 9. Eluted proteins were separated on a 13% SDS-PAGE gel. The gel was stained overnight with InstantBlue (Abcam), lanes were excised, and in-gel digested with trypsin. LC-MS/MS analysis was performed using linear 60-min gradients and an Easy-nLC 1200 system coupled to a Q Exactive HF mass spectrometer (Thermo Fisher Scientific, Germany) by the Proteome Center at the University of Tübingen. Co-IPs with SR834 and DR825 as control were performed identically to α-GFP-FraD co-IPs.

Detected proteins with at least 100 times increased abundancy in the sample compared to the control were analyzed on their taxonomic distribution using STRING version 11[55] and HmmerWeb version 2.41.1. Proteins were considered as candidates for FraD-interacting proteins, if they showed a similar taxonomic distribution as FraD, which means conservation in filamentous cyanobacteria.

## In silico protein and gene analysis

Protein secondary structure prediction was performed using Protter webinterface version 1.0[36]. InterPro 82.0[35] and Pfam 33.1[37] were used to scan for conserved domains.

## Light- and fluorescence microscopy

A Leica DM2500 B microscope with a Leica DFC420C camera was used for light microscopy. Fluorescence microscopy was performed with a 100x/1.3 oil objective lens of a Leica DM5500 B microscope connected to a Leica DFC360FX camera. GFP and chlorophyll fluorescence were excited using a BP470/40 nm or BP535/50 nm filter and emission was monitored using a BP525/50 nm or BP610/75 nm filter, respectively. Z-stacks with 0.1 μm intervals were taken to perform three-dimensional deconvolution using the built-in function of the Leica ASF software.

For quantification of the fluorescence in the septum, each xy pixel of 20 slices of a raw (not deconvoluted) z-stack was summed into one image and the mean intensity of septal ROIs was measured. Background fluorescence was measured with identical ROIs in the cytoplasm. The averaged background fluorescence was subtracted from every single septum measurement and used for normalization of the values.

To visualize the presence of the heterocyst polysaccharide layer, the cells were stained with Alican blue as described[56].

## Immunolocalization of FraD

Immunolocalization of FraD in *Nostoc* strains was principally performed as described by Büttner et al. with some minor changes[57]. *Nostoc* filaments were grown for 3 days on BG11 plates and resuspended in 1 mL PBS to $OD_{750} = 1$. After three washing steps, the cells were fixed with 1 mL of HistoChoice Tissue Fixative (Sigma-Aldrich) for 10 min at RT and 30 min at 4 °C. Cells were washed three times with PBS, once with 70% EtOH prechilled at −20 °C and again with PBS. Next, cells were treated with 1 mg/mL lysozyme in 1 mL GTE buffer (50 mM glucose, 20 mM Tris-HCl pH 7.5, 10 mM EDTA) for 5 min at RT, resuspended in 200 μL PBS and dropped onto a Polysine slide (ThermoFisher Scientific, Germany). Cells were dried at RT and rehydrated with 200 μL PBS for 5 min. After blocking with 200 μL 2% w/v BSA in PBS for 20 min, cells were incubated overnight in a wet chamber at 4 °C with 10 μg/mL α-FraD antibodies (see above) in BSA-PBS. Cells were washed five times with PBS and incubated 2 h at RT in the dark with FITC-coupled α-rabbit antibodies (1:200 in BSA-PBS, Sigma-Aldrich). After washing and drying, one drop of Vectashield Mounting Medium H-1200 (Vector Laboratories, USA) was applied, covered with a coverslip and sealed with nail polish. Fluorescence was imaged using a DM5500B Leica microscope and a DFC360FX monochrome camera. Autofluorescence was detected as described before, FITC-fluorescence was detected by using a BP470/40 nm excitation filter and a BP525/50 nm emission filter. Z-stacks with 0.1 μm intervals were taken and applied to the Leica ASF built-in function for 3D-deconvolution. Validation of the antibody was achieved by using the Δ*fraD* mutant as control.

## Transmission electron microscopy of heterocyst

For transmission electron microscopy analysis of heterocysts, ultra-thin sections were prepared from cells incubated for 2 days on BG11₀ agar plates void of nitrate. Sample preparation was performed as described in ref. 58. Prior sectioning, the cells were fixed with glutaraldehyde and potassium permanganate, the ultra-thin sections stained with uranyl acetate and lead citrate. The samples were examined with a Philips Tecnai10 microscope at 80 kV.

## FRAP, CCCP-FRAP and UV-FRAP

For FRAP and CCCP-FRAP experiments[17], 500 μL cells were stained with 8 μL of a calcein acetoxymethylester solution (1 mg/mL in DMSO) and incubated for 90 min the dark at 28 °C. After three washing steps with BG11 medium, cells were incubated either with (50 μM CCCP) or without protonophore for further 90 min in the dark. Onto a BG11 agar plate 10 μL of the stained cells were spotted for imaging with excitation at 488 nm of a laser at 0.2% intensity of a Zeiss LSM 800 confocal microscope using a 63x/1.4 oil-immersion objective and the ZEN 2.3 or 2.6 (blue edition) software. Calcein fluorescence emission was detected at 400–530 nm simultaneously with chlorophyll autofluorescence emission at 650–700 nm. Bleaching of a specific cell was achieved by shortly increasing the laser intensity to 3.5% after 5 pre-bleached images were taken. Recovery of fluorescence in the bleached cell was monitored in 1 s intervals for 30–180 s. Analysis of the acquired images was done with ImageJ (version 1.51j) and GraphPad Prism 6 as described earlier[17]. UV-FRAP was performed with slight modifications. After calcein staining and washing of 3-day old cultures, 10 μL of cells were

spotted onto agar and treated with UV light inside of Bio-Link Cross-linker BLX-E254 with six 8 W lamps for 2 min. FRAP response was measured as described above.

## Isolation of septal peptidoglycan and transmission electron microscopy

Septal PG was isolated after Kühner et al.[59] with some changes. Cells grown for 3 days on BG11 agar plates were resuspended in 700 μL 0.1 M Tris-HCl pH 6.8 and sonicated (Branson Sonifier 250) for 2 min at a duty cycle of 50% and output control 1. After addition of 300 μL 10% SDS, the samples were boiled for 30 min at 99 °C with shaking at 300 rpm. The suspension was washed two times with ddH$_2$O (5 min, 10,000 rpm), incubated for 30 min in a sonifier waterbath. After washing with ddH$_2$O, the debris was resuspended in 1 mL 50 mM Na$_3$PO$_4$ pH 6.8 and incubated 3 h at 37 °C with 300 μg α-Chymotrypsin under constant rotation. After that, α-Chymotrypsin was added again, and the samples were incubated over night at 37 °C. The next day, the enzyme was heat inactivated for 3 min at 99 °C. After another sonication (30–90 s) and washing step with ddH$_2$O, the PG septa were resuspended in 50–500 μL ddH$_2$O. All solutions were filtered with a 0.22 μm filter.

10 μL of isolated septal PG suspension was incubated on an UV-irradiated (16 h) formvar/carbon film-coated copper grid (Science Services GmbH, Munich, Germany) for 10–30 min. Samples were stained with 1% (w/v) uranyl acetate and imaged with a Philips Tecnai10 electron microscope at 80 kV equipped with a Rio Camera (Gatan).

## Plunge freezing of *Nostoc* cells

*Nostoc* cultures were concentrated by gentle centrifugation (1000 × *g*, 5 min) and removing 2/3 of the medium. 3.5 μL of cell suspension was applied on glow-discharged copper or gold EM grids (R2/2, Quantifoil) and automatically back-blotted (Weiss et al.[60]) for 4–6 s, at 22 °C with 100% humidity and plunged into liquid ethane/propane[61] using a Vitrobot plunge freezing robot (ThermoFisher Scientific, Waltham, MA). Frozen grids were stored in liquid nitrogen.

## Cryo-focused ion beam milling

Thin lamellae through plunge frozen *Nostoc* filaments were obtained by automated sequential FIB milling according to[62]. Briefly, EM grids were clipped into FIB milling autoloader-grids (ThermoFisher Scientific, Waltham, MA) and mounted onto a 40° pre-tilted grid holder[63] (Leica Microsystems GmbH, Vienna, Austria) using a VCM loading station (Leica Microsystems GmbH, Vienna, Austria). For all grid transfers under cryo-conditions, a VCT500 cryo-transfer system (Leica Microsystems GmbH, Vienna, Austria) was used. Grids were sputter-coated with a ~4 nm thick layer of tungsten using ACE600 cryo-sputter coater (Leica Microsystems GmbH, Vienna, Austria) and afterwards transferred into a Crossbeam 550 FIB-SEM dual-beam instrument (Carl Zeiss Microscopy, Oberkochen) equipped with a copper-band cooled mechanical cryo-stage (Leica Microsystems GmbH, Vienna, Austria). The gas injection system (GIS) was used to deposit an organometallic platinum precursor layer onto each grid. Grid quality assessment and targeting of the cells were done by scanning EM (SEM) imaging (3–5 kV, 58 pA). Coordinates of chosen targets were saved in the stage navigator and milling patterns were placed onto targets' FIB image (30 kV, 20 pA) using the SmartFIB software. To mill 11 μm wide and ~200 nm thick lamellae a total of four currents were used and currents were gradually reduced according to lamella thickness [rough milling (700 pA, 300 pA, and 100 pA) and polishing (50 pA)]. After the milling session, the holder was brought back to the loading station, grids were unloaded and stored in liquid nitrogen.

## Cryo-electron tomography

Lamellae through *Nostoc* filaments were examined by cryo-electron tomography. Data were collected on a Titan Krios 300 kV FEG

transmission electron microscope (ThermoFisher) using a Quantum LS imaging filter (slit width 20 eV) and K2 Summit direct electron detector (Gatan). A low magnification (135x) overview of the grid was recorded using SerialEM[64] to identify lamellae. Bi-directional tilt series were collected automatically using a custom-made SerialEM script. Target tracking was achieved via cross-correlation to the previous record image. For focusing, a single autofocus routine was performed at zero degree and focus change over the entire tilt series was estimated using the focus equation from UCSF tomography[65].

Tilt series covered an angular range from −50° to +70° at 2° increments with a defocus of −8 μm, total accumulated dose of -120–140 e⁻/Å² and a pixel size of 3.45 Å. Tomogram reconstruction was performed according to[60] using IMOD package[66]. A deconvolution filter[67] was used for tomograms shown in figures.

## Subtomogram averaging

Subtomogram averaging was performed using Dynamo[68]. Briefly, SJs were identified visually in individual tomograms and their coordinates were saved as oriented particles models in the catalog. For the initial averages, 418, 282, 89, 118 and 369 particles were picked for WT (open state), WT (closed state), *sepN⁻* mutant, *sepN-sfgfp* and *sepN-mbp-sfgfp* from 22, 18, 18, 11 and 33 tomograms, respectively. To exclude reference bias, first templates were created by averaging fifty randomly picked particles for each dataset. Particles were split in half for subsequent FSC calculations and aligned for three iterations using 2 × 2-binned tomograms [box-size 72 × 72 × 72 pixels, alignment mask: sphere (radius = 35 pixels)] for WT (open state), WT (closed state), *sepN⁻* mutant datasets. For *sepN-sfgfp* and *sepN-mbp-sfgfp* datasets, 4 × 4-binned tomograms were used [box-size 44 × 44 × 44 pixels, alignment mask: sphere (radius = 21 pixels)]. To attenuate a prominent missing wedge in the average due to preferential particle orientation, we randomized the rotational angle using dynamo_table_randomize_azimuth(table) matlab command. Another round of 4 iterations was performed and obtained averages and tables were used for refinement. Due to our observations in the unsymmetrized average and according to results in[17], we applied five-fold symmetry for WT (open state), WT (closed state), *sepN⁻* mutant and *sepN-sfgfp* averages during the final refinement run (9 iterations with a tight alignment mask). No symmetry was applied for the *sepN-mbp-sfgfp* average. The final, cross-correlation cleaned (CC-cutoff: 0.18) averages resulted from 387 (WT open state), 248 (WT closed), 87 (*sepN⁻* mutant), 116 (*sepN-sfgfp)* and 334 (*sepN-mbp-sfgfp*) particles. To estimate the resolution of each dataset, a fourier shell correlation was calculated from their half-maps using dynamo. The difference map between subtomogram averages of WT closed state and *sepN⁻* mutant was generated with the diffmap program (https://grigoriefflab.umassmed.edu/diffmap) using averages that were low-pass filtered to a resolution of 35 Å. 3D rendering and coloring of different SJ modules were done with Chimera.

## UV-treatment and light microscopy

*Nostoc* cultures were grown on agar plates and resuspended in BG11 medium. OD$_{750}$ was adjusted to ~0.8 and 10 ml of the cultures were transferred into a petri dish. Cultures were treated with two Sankyo Denki Germicidal 68 T5 UV-C lamps for 5 mins in a repurposed Herolab UV DNA Crosslinker CL-1. Ten μl of *Nostoc* cultures were spotted on a BG11 agar plate and left until the liquid was fully absorbed. An area was cut out and mounted on a cell culture imaging dish. Time-lapse movies were recorded using a 40x objective on a Leica Thunder Imager 3D Cell Culture equipped with a Leica DFC9000 GTC CMOS camera (2048 × 2048 pixels, pixel size 6.5 μm). Images were recorded every 60 s over a time course of 5 h. To analyze the survival rate after UV-treatment on culture level, 3 ml of *Nostoc* cultures were treated 1 min with UV light and a dilution series was spotted on a pre-warmed BG11 plate. The plate

was incubated in the dark for 24 h at 28 °C. Afterwards, incubation was continued for 10 days with continuous light.

### Test on significance

Statistical analysis was performed with GraphPad Prism version 6.01. Statistical significance of two groups was tested using an unpaired Student's $t$-test. Comparison of one group with multiple other groups was performed via ordinary one-way ANOVA followed by Dunnett's or Tukey's multiple comparison test. Significance $P$ is indicated with asterisks: $*P \leq 0.05$; $**P \leq 0.01$; $***P \leq 0.001$; $****P \leq 0.0001$; ns (not significant): $P > 0.05$.

### Reporting summary

Further information on research design is available in the Nature Portfolio Reporting Summary linked to this article.

## Data availability

Subtomogram averages and representative cryo-tomograms generated in this study have been deposited in the EMDB database (https://www.ebi.ac.uk/emdb/) under accession codes: EMD-16012, EMD-16033, EMD-16034, EMD-16053, EMD-16054, EMD-16056, and EMD16061. Source data are provided with this paper. All other data that support the findings of this study are available from the corresponding author (I.M.) upon request. Source data are provided with this paper.

## Code availability

The custom-made SerialEM script for acquisition of cryo-electron tomograms is available upon request.

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

## Acknowledgements

We thank Mirita Franz-Wachtel of the Proteome Center Tübingen for performing LC-MS/MS, and Jan Bornikoel for construction of the C.K3t4 cassette. We thank Karl Forchhammer for fruitful discussions. Furthermore, we thank Teresa Müller for performing the FRAP of DR825.848, Claudia Menzel for technical assistance, Enrique Flores for providing strain CSVT2 and Peter Wolk for pRL-plasmids. Work in Tübingen was supported by the German research foundation (GRK1708 and DFG-MA1359/7). We acknowledge instrument access at the imaging platform ScopeM at ETH Zürich. We acknowledge the Tübingen Structural Microscopy Core Facility (funded by the Excellence Strategy of the German Federal and State Governments) for their support & assistance. Work in Zurich was supported by the Boehringer Ingelheim Fonds and the NOMIS foundation.

## Author contributions

A.-K.K. designed and performed experiments, analyzed and interpreted data, drafted the work and wrote the manuscript. P.T. designed and

performed cryoET experiments, analyzed and interpreted the corresponding data, revised the manuscript critically. A.J. designed and performed experiments, analyzed and interpreted the corresponding data, revised the manuscript critically. M.P. designed and supervised the research in Zürich, interpreted data, revised the manuscript critically. G.L.W. designed, performed and supervised the research in Zürich, interpreted data and wrote the manuscript. I.M. designed and supervised the research in Tübingen, interpreted data and wrote the manuscript. All authors approved the final manuscript.

## Funding

## Competing interests
The authors declare no competing interests.
