## [Peer Review File · Nature Communications]

SepN is a septal junction component required for gated cell-cell communication in the filamentous cyanobacterium *Nostoc*Reviewer #1 (Remarks to the Author):

The manuscript by Kieninger et al. describe the identification and characterization of SepN, involved in the assembly of septal junctions (SJs) in filamentous cyanobacteria. Filamentous cyanobacteria, in particular those able to form nitrogen fixations cells, heterocysts, are multicellular organisms, relying on intercellular exchanges of nutrients and signals for growth. The cells on a filament are connected by a common outer membrane, and crosslinked septal peptidoglycan (PG). Nanopores are drilled by amidases through the septal PG, and through which a protein complex called a septal junction, serves as the structure for intercellular molecular exchange. The SJ can be reversibly opened or closed in response to stress or diazotrophic conditions, known as the gating mechanism.

Although the function of SJs have been extensively studied with a dozen of genes known to be involved in their establishment or functioning, mostly by molecular genetics, neither their structural components nor their assembly process is well known. Therefore, the new approach, name Co-IP, the authors used to identify a new payer in these processes, and the multiple approaches (genetics, cell biology, in-situ tomography etc), to characterize SepN, represent a significant advance for our understanding on a prokaryotic gating apparatus for cell-cell communication. I have some comments to help the authors to revise their manuscript and clarify some important points.

- 1) Lines 62-65; lines 580-583 and elsewhere. About the function of FraD. I don't think that we have direct evidence to show that FraD is a structural component of SJs. As shown in a previous publication, the plug and cap modules of SJs are missing in the fraD mutant. The corresponding protein may, or may not be the real component of the plug, as proposed. It remains anyway a proposal, and direct evidence is still lacking. The images obtained by in-situ tomography is not enough in resolution to know the nature of each component. FraD, as well as SepN, could be a structural component of, or just a chaperon involved in the assembly of, SJs. Lines 636-639, the conclusion is challenged by the data of the authors.
- 2) The same can be said for SepN. The authors should just propose that it is a protein involved directly, or indirectly in the formations of SJs.
- 3) The authors used Co-IP, to identify SepN. They have very robust data for protein-protein interaction in vivo. Do remember that these data cannot distinguish whether FraD-SepN interaction is direct, or mediated through another component. It would be interesting to use an alternative method to determine the interaction of these two, for example, by yeast or bacterial two hybrid system.
- 4) Lines 36-38, filamentous cyanobacteria, even without cell differentiation, have also the multicellular behaviours.
- 5) Lines 344-346; lines 572-574. Please show the nature of the major hits, as a Table. These data will be very useful for the community.
- 6) Lines 391-394. sepN is a putative membrane protein. How to explain its localization pattern in heterocysts, surrounding the polar granules?
- 7) About sepN mutant phenotype, and Fig. 2. Please show, or describe the data about filament integrity, and diazotrophic growth. In the discussion (lines 634-636), it looks like that there is no particular phenotypes, but these results should be shown. It should also be explained that if the SJs in the sepN mutant are in a closed state, it must impair cell-cell communication, thus the ability of the mutant to grow diazotrophically.
- 8) SepN-GFP is partially functional. Have the authors tried to use GFP to different parts of SepN? Is there a linker between GFP and SepN inserted?
- 9) Lines 449-450. Nanopore cannot be the scaffold of, or build the scaffold for SJ assembly. It is necessary for the complex to traverse the cell wall in between, and connect the cells together through SJs. At least, this expression is misleading.
- 10) Line 458. Say sepN mutant, instead of WT lacking sepN.
- 11) Figure 4, panel D. Please tell which figure corresponds to what.

Reviewer #2 (Remarks to the Author):

In this well written study, Kieninger describe the discovery of a new protein component of septal junctions in the cyanobacterium Nostoc. They go on to show that this protein (sepN) is not important for the making of these junctions but for their activity.

My main conclusion is that this is a perfectly nice molecular bacterial study, but for a selective audience, namely those working on septal junctions in cyanobacteria.

Minor remarks. The Discussion is a repetition of the results, and it might be easier to use a Result & Discussion format and end with a Conclusion.

I could only detect 3 minor typo's:

Line 448; set out to

Line 572; further evidence that (no comma)

Line 575; Despite FraC being encoded in

Reviewer #3 (Remarks to the Author):

The manuscript by A.-K. Kieninger et al. aims at revealing the molecular architecture of septal junctions in the cyanobacterium Nostoc sp. using molecular biology/biochemistry techniques and cryo-electron tomography. Septal junctions are multiprotein assemblies mediating cell-cell communication. As such they are an interesting and challenging target for structural studies.

Using immunoprecipitation, the authors identified a new protein component of septal junctions, SepN, which interacts with the previously identified FraD which served as a bait. SepN locates to septal junctions, but mutants showed normal septal nanopore formation. Possibly, SepN is part of the plug serving to close septal junctions.

The authors used cryo-electron tomography to obtain an insight into the molecular architecture of septal junctions. Although they used state-of-the art methodology and tools the results fall short of a detailed and definitive elucidation of this structure. It seems to this reviewer that septal junctions are not exactly deterministic structures and this causes problems in subtomogram averaging. Moreover, the numbers of subtomograms available for averaging (and possibly classification) is relatively small. All this results in a rather low resolution which is not good enough for any kind of integrative model building. Moreover, I am concerned that imposing five-fold symmetry on the whole assembly could yield misleading results. There is no guarantee e. g. that the plug follows the symmetry of the septum. At least the authors should show the non-symmetrized averages.

Taken together, I think this is a good first step but improvements are needed to make it Nature Communications material.

Response to Referees of manuscript **NCOMMS-22-03312** with the adjusted title

“The septal junction subunit SepN is required for gated cell-cell communication”.

Reviewer #1 (Remarks to the Author):

The manuscript by Kieninger et al. describe the identification and characterization of SepN, involved in the assembly of septal junctions (SJs) in filamentous cyanobacteria. Filamentous cyanobacteria, in particular those able to form nitrogen fixations cells, heterocysts, are multicellular organisms, relying on intercellular exchanges of nutrients and signals for growth. The cells on a filament are connected by a common outer membrane, and crosslinked septal peptidoglycan (PG). Nanopores are drilled by amidases through the septal PG, and through which a protein complex called a septal junction, serves as the structure for intercellular molecular exchange. The SJ can be reversibly opened or closed in response to stress or diazotrophic conditions, known as the gating mechanism.

Although the function of SJs have been extensively studied with a dozen of genes known to be involved in their establishment or functioning, mostly by molecular genetics, neither their structural components nor their assembly process is well known. Therefore, the new approach, name Co-IP, the authors used to identify a new player in these processes, and the multiple approaches (genetics, cell biology, in-situ tomography etc), to characterize SepN, represent a significant advance for our understanding on a prokaryotic gating apparatus for cell-cell communication. I have some comments to help the authors to revise their manuscript and clarify some important points.

We thank the reviewer for carefully evaluating our manuscript and for the constructive feedback below.

1) Lines 62-65; lines 580-583 and elsewhere. About the function of FraD. I don't think that we have direct evidence to show that FraD is a structural component of SJs. As shown in a previous publication, the plug and cap modules of SJs are missing in the fraD mutant. The corresponding protein may, or may not be the real component of the plug, as proposed. It remains any way a proposal, and direct evidence is still lacking. The images obtained by in-situ tomography is not enough in resolution to know the nature of each component. FraD, as well as SepN, could be a structural component of, or just a chaperon involved in the assembly of, SJs. Lines 636-639, the conclusion is challenged by the data of the authors.

We agree with the reviewer that at the current resolution of our structural data, it is not possible to conclude a direct structural role of FraD in forming the plug module. In the previous publication mentioned by the reviewer (Weiss et al 2019), we imaged a GFP-FraD expressing mutant with cryo-electron tomography (cryo-ET), which visualized an additional density in the lumen of the septal junction (SJ) tube, directly adjacent to the plug. A difference map that was generated using the wild type and GFP-FraD SJ subtomogram averages identified the density in the tube lumen as the only major difference. The observed density therefore likely represents the GFP-tag fused to FraD. The data suggests that FraD is a structural element of septal junctions, with its N-terminus facing the tube lumen and localizing near the plug module. Given our new data (see response to point 2 below), we concluded that FraD might serve as a linker between the plug, cap, and cytoplasmic membrane, rather than forming the plug module itself. Accordingly, we changed the statements on FraD (now lines 55-56, 304-308).

2) The same can be said for SepN. The authors should just propose that it is a protein involved directly, or indirectly in the formations of SJs.

To further provide evidence that SepN is indeed a structural subunit of SJs, we fused one copy of maltose binding protein to SepN-sfGFP. This increased the size of the tag to SepN, allowing its visualization with cryo-ET. First, we verified that SepN-MBP-sfGFP still localizes to the septum by using fluorescent light microscopy (new Fig. 6a). Next, we acquired cryo-tomograms of cryo-focused ion beam-thinned cells and observed aberrant SJ architecture. While the cap module was always missing ($n=33$ tomograms, 369 SJ ends), a plug module was still visible and strikingly, we could observe additional densities adjacent to the plug (new Fig. 6b,c). A subtomogram average of these SJs ends further showed that the additional densities were connected to the plug module and pointing to the cytoplasm (new Fig. 6d). Even though it is very likely that the observed densities represent the MBP-sfGFP tag and therefore indicating a structural role of SepN in the plug module, we cannot completely rule out a regulatory role of SepN in the correct assembly of the cap and plug module, which might be impaired due to the MBP-tag. We discuss this now in line 261-265 and in the conclusion line 297-299.

3) The authors used Co-IP, to identify SepN. They have very robust data for protein-protein interaction *in vivo*. Do remember that these data cannot distinguish whether FraD-SepN interaction is direct, or mediated through another component. It would be interesting to use an alternative method to determine the interaction for these two, for example, by yeast or bacterial two hybrid system.

We thank the reviewer for the suggestion. However, we think that our data already suggest a direct interaction between FraD and SepN, since SepN was the only protein that was among the highest hits in all performed Co-IPs. If there would be any interaction mediating protein involved to link FraD and SepN, we would assume to have identified this protein as a major hit in our Co-IP experiments.

To further examine the nature of the interaction between FraD and SepN, we performed bacterial two hybrid experiments (BACTH) as suggested by the reviewer. For this, the protocol published by Battesti and Bouveret was followed (Battesti & Bouveret 2012; <https://doi.org/10.1016/j.ymeth.2012.07.018>). The T25 part of the adenylate cyclase domain was cloned upstream of the N-terminus of *alr2393* (*fraD*). The counterpart, T18, was cloned 5' of *all4109* (*sepN*). Unfortunately, no interaction was observed in *E. coli* on LB agar plates supplemented with X-Gal or on MacConkey plates containing lactose, after incubation for 1-3 days at 20 °C or 28 °C. Next, we tried to fuse the adenylate cyclase parts to the soluble C-terminal domains of both, FraD and SepN (excluding the membrane domains), assuming expression of these constructs in the cytoplasm. Also here, we did not get any positive interaction signal. As further yet also negative attempt, we repeated the experiments with codon-optimized *fraD* and *sepN* sequences for *E. coli*.

Although we did not find any positive interaction signal via BACTH, it does not exclude a direct FraD-SepN interaction. Since *E. coli* is not the native host, these proteins could be unstable or misfolded. The intracellular conditions in *E. coli* might differ substantially from the native environment of FraD and SepN in *Nostoc*. This becomes especially apparent considering the complex nature of septal junctions, which span the septal peptidoglycan, two cytoplasmic membranes and exhibit a cytoplasmic subunit. The stability of protein complexes outside their defined native intracellular environment appears to be a common issue in BACTH attempts, even more pronounced if the complex of interest involves a membrane protein or is spanning different compartments (Oikonomou & Jensen, 2017).

In addition to BACTH, we performed *in vitro* protein-protein interaction studies, using heterologously expressed and purified proteins. The periplasmic domains of FraD and SepN were fused to different functional tags and expressed in *E. coli*. Unfortunately, the yields after purification were very low, caused by either polymerization into macromolecules or by aggregation of misfolded or unstable proteins. Nevertheless, we attempted *in vitro* interaction studies (batch pulldowns and biolayer interferometry), but did not obtain clear results.

4) Lines 36-38, filamentous cyanobacteria, even without cell differentiation, have also the multicellular behaviours.

Thank you for pointing this out. We changed the sentence accordingly. Lines 36-37.

5) Lines 344-346; lines 572-574. Please show the nature of the major hits, as a Table. These data will be very useful for the community.

A detailed list with all hits can now be found in Supplementary Data 1.

6) Lines 391-394. *sepN* is a putative membrane protein. How to explain its localization pattern in heterocysts, surrounding the polar granules?

Heterocysts differentiate from vegetative cells under extensive genetical, metabolic and morphological alterations. This includes restriction of the septum and reorganization of intracellular membranes. Membrane proteins residing in the broader septum of the former vegetative cell have to move with the cytoplasmic membrane during the transition of the septum to a narrow “polar neck”. The faint fluorescence surrounding the polar granules could be left-over proteins of this constriction. Similar observations were made for SepJ that is involved in nanopore formation and intercellular communication. Immunogold labeling of FraG/SepJ in the heterocyst neck showed particles not only at the septum but dispersed around the polar granules (Omairi-Nasser et al 2015, Figure 4; <https://doi.org/10.1073/pnas.1512232112>). In the septa between vegetative cells, however, SepJ-GFP localizes exclusively in the center of the septum.

7) About *sepN* mutant phenotype, and Fig. 2. Please show, or describe the data about filament integrity, and diazotrophic growth. In the discussion (lines 634-636), it looks like that there is no particular phenotypes, but these results should be shown.

We thank the reviewer for this suggestion and included data on diazotrophic growth and heterocyst formation/architecture of the *sepN*⁻ mutant (shown in Fig. 3a-c and Supplementary Fig. S2 and S3). Whereas no differences between the *sepN*⁻ mutant and the WT could be detected in conditions with nitrate, the growth of the mutant was slowed down after 2 days under nitrogen limiting conditions (Fig. 3b) and the filament length was slightly reduced during diazotrophic growth (new Supplementary Fig. 3). However, this phenotype was clearly different from the Δ *fraD* mutant, which heavily fragments under these conditions. Our data implies that the depletion of SepN has only minor effects on diazotrophic growth. Data is discussed in lines 169-170, 301-302, 308-310.

It should also be explained that if the SJs in the *sepN* mutant are in a closed state, it must impair cell-cell communication, thus the ability of the mutant to grow diazotrophically.

It is correct that the SJ cap module in the *sepN*⁻ mutant is in the closed conformation, but importantly, the plug module is absent in these SJs. Our FRAP data indicates that the intercellular communication in the *sepN*⁻ mutant is reduced compared to WT, but still occurring at a lower rate. This indicates that the plug module is required for complete closure of SJs between vegetative cells. Furthermore, the *sepN*⁻ mutant was unable to gate communication between vegetative cells after ionophore treatment.

Interestingly, we did not observe a major phenotype of the *sepN*⁻ mutant in diazotrophic growth or heterocyst differentiation (see above). This stands in contrast to the phenotype observed for the Δ *fraD* mutant (Merino-Purto et al. 2010, doi:10.1111/j.1365-2958.2009.07031.x). It remains unknown why

FraD and not SepN is required for heterocyst function and it needs further investigation of the heterocyst – vegetative cell communication apparatus to elucidate the individual role of SJ modules during heterocyst differentiation. We discuss this now in lines 301-302 and 308-310.

8) SepN-GFP is partially functional. Have the authors tried to use GFP to different parts of SepN? Is there a linker between GFP and SepN inserted?

The here described GFP-fusion to SepN included a 5x-GS linker between sfGFP and the N-terminus of SepN. Due to a clear septal localization signal for SepN and the ability of the *sepN-sfgfp* mutant to gate intercellular communication, we do not see the necessity to fuse GFP to different parts of SepN. The reduced fluorescence recovery rate R observed for the *sepN-sfgfp* mutant might be a hint on steric hinderance of diffusion of molecules through SJs by GFP, similar to the recently described GFP-FraD mutant (Weiss et al 2019). Furthermore, we acquired additional cryo-ET data on the *sepN-sfgfp* mutant and a subtomogram average of SJs did not reveal major differences in SJ architecture compared to the WT open state (new Supplementary Figure 7).

9) Lines 449-450. Nanopore cannot be the scaffold of, or bulid the scafld for SJ assembly. It is necessary for the complex to travers the cell wall in between, and connet the cells together through SJs. At least, this expression is misleading.

We agree with the reviewer and deleted the sentence. In the introduction we describe it accordingly (lines 48-50)

10) Line 458. Say sepN mutant, instead of WT lacking sepN.

We changed the manuscript accordingly.

11) Figure 4, panel D. Please tell which figure corresponds to what.

Thank you for pointing this out. We added another panel to avoid confusion. Now Figure 5d,e.

Reviewer #2 (Remarks to the Author):

In this well written study, Kieningner describe the discovery of a new protein component of septal junctions in the cyanobacterium *Nostoc*. They go on to show that this protein (sepN) is not important for the making of these junctions but for their activity.

My main conclusion is that this is a perfectly nice molecular bacterial study, but for a selective audience, namely those working on septal junctions in cyanobacteria.

We thank the reviewer for the positive evaluation. We believe that our manuscript is of interest for the broad readership of *Nature Communication*, since it substantially adds to our understanding of the evolution of multicellularity and cell-cell communication.

The branching of the bacterial order of *Nostocales* was estimated to date back more than two billion years ago (e.g., Schirromeister et al., 2013, www.pnas.org/cgi/doi/10.1073/pnas.1209927110). Septal junctions therefore predate metazoan gap junctions by more than a billion years, which makes them an ancient cell-cell junction and paved the way for early evolution of a multicellular lifestyle. The convergent evolution of gated cellular communication in such divergent lineages emphasizes the importance of controlling cellular communication across the different domains of life, especially under stress conditions. Strikingly, we now show in the revised manuscript that gated cell-cell communication is crucial for the survival of this multicellular cyanobacterium after environmental stress (new Figure 7, new Supplementary Figure 8, and in the text lines 272-295). Gated septal junctions likely prevent the leakage of cytoplasmic components into damaged cells, which ensures the survival of the rest of a *Nostoc* filament.

Besides an evolutionary aspect, cyanobacteria are among the most abundant organisms on our planet, being a major player in the global carbon and nitrogen cycle. Understanding how multicellular cyanobacteria control communication and deal with environmental stress is therefore of broad interest, especially in times of global warming.

Minor remarks. The Discussion is a repetition of the results, and it might be easier to use a Result & Discussion format and end with a Conclusion.

We thank the reviewer for the suggestion and adapted the manuscript accordingly.

I could only detect 3 minor typo's:

Line 448; set out to

Changed, now line 206

Line 572; further evidence that (no comma)

This sentence was deleted during revision.

Line 575; Despite FraC being encoded in

This sentence was deleted during revision.

Reviewer #3 (Remarks to the Author):

The manuscript by A.-K. Kieninger et al. aims at revealing the molecular architecture of septal junctions in the cyanobacterium *Nostoc* sp. using molecular biology/biochemistry techniques and cryo-electron tomography. Septal junctions are multiprotein assemblies mediating cell-cell communication. As such they are an interesting and challenging target for structural studies.

Using immunoprecipitation, the authors identified a new protein component of septal junctions, SepN, which interacts with the previously identified FraD which served as a bait. SepN localizes to septal junctions, but mutants showed normal septal nanopore formation. Possibly, SepN is part of the plug serving to close septal junctions.

We appreciate that the reviewer took the time to carefully evaluate our manuscript and we thank him/her for the feedback and comments below.

The authors used cryo-electron tomography to obtain an insight into the molecular architecture of septal junctions. Although they used state-of-the-art methodology and tools the results fall short of a detailed and definitive elucidation of this structure. It seems to this reviewer that septal junctions are not exactly deterministic structures and this causes problems in subtomogram averaging. Moreover, the numbers of subtomograms available for averaging (and possibly classification) is relatively small. All this results in a rather low resolution which is not good enough for any kind of integrative model building.

We agree with the reviewer that the achieved resolution in our subtomogram averages is not sufficient to perform integrative model building, but this was never attempted in this manuscript. The reasons behind the limited resolution are the intrinsic flexibility of SJs in adapting to the curvature of the cytoplasmic membrane as well as rather low particle numbers per septum. But importantly, the major insight gained from the cryo-tomograms and from the subtomogram averages were (1) the absence of the SJ plug module in the *sepN*⁻ mutant and (2) that the cap module of the *sepN*⁻ mutant revealed similarities to the cap in the closed state of WT after CCCP treatment. We believe that our achieved resolution is sufficient to make these statements.

To achieve a better resolution for potential integrative model building, it would be necessary to increase the particle number by at least ~100 times. Counterproductively, we noticed a reduced number of SJs per septum in the *sepN*⁻ mutant (on average 4.9 SJs / tomogram; $n = 18$ tomograms with SJs, total 31 tomograms) compared to WT (on average 19 SJs / tomogram) (numbers now stated in the manuscript, line 223, 227-231). This means that only a very limited number of particles can be picked per septum/tomogram. Due to the fact that all cells need to be thinned by cryo-focused ion beam milling prior tomogram acquisition, this would implement an extraordinary time-consuming approach, beyond the scope of this paper.

Nevertheless, to gain more insights into the localization of SepN within the multi-protein architecture of SJs, we believed it is more insightful to acquire data on different mutant strains. First, we collected new cryo-tomograms of the *sepN-sfGFP* mutant. Other studies already successfully visualized GFP as additional densities in subtomogram averages (including the SJ protein FraD). However, subtomogram averages of SJs from the *sepN-sfGFP* mutant did not reveal any additional densities caused by GFP (now discussed in line 249-253 and in new Supplementary Fig. 7). The absence of an additional density in a subtomogram average can be explained by a potential flexibility of the sfGFP-tag in relation to the SJ modules.

By adding an additional maltose binding protein (MBP) on top to the sfGFP-fusion construct, we further increased the size of the tag to SepN (Supplementary Fig. 1). First, we verified that SepN-MBP-sfGFP still localizes to the septum by using fLM. Next, we acquired tomograms of FIB-milled cells and observed

aberrant SJ architecture. While the cap module was always missing (n= 33 tomograms, 369 SJ ends), a plug module was still visible and strikingly, we could observe an additional density adjacent to the plug (new Fig. 6b,c). A subtomogram average of these SJs ends further showed that the additional densities were connected to the plug (new Fig. 6d). It is likely that the observed densities represent the MBP-sfGFP tag and therefore indicating a structural role of SepN in the plug module. The large size of the potential MBP-tag might also explain the absence of a cap module due to sterical hindrance. To our knowledge, this is the first study which used a MBP tag to localize proteins in cryoET data.

Moreover, I am concerned that imposing five-fold symmetry on the whole assembly could yield misleading results. There is no guarantee e. g. that the plug follows the symmetry of the septum. At least the authors should show the non-symmetrized averages.

We agree with the reviewer and thank him/her for the suggestion to include the unsymmetrized version of our subtomogram averages (new Supplementary Fig. 5). Already in the unsymmetrized subtomogram average, a 5-fold rotational symmetry in both, the cap *and* plug is detectable in the WT open state. The symmetry of the plug was not observed in the previous publication of Weiss et al. and represents a new finding of this manuscript.

Taken together, I think this is a good first step but improvements are needed to make it Nature Communications material.

To further show the importance of understanding the underlying principles of a controlled cellular communication in multicellular organisms, we analyzed individual cell lysis events and the survival rate of *Nostoc* after environmental stress. We could show that after treatment with UV-light, cellular communication is aborted in the WT, which potentially prevents the leakage of cytoplasmic components into damaged cells. This enables the survival of the rest of the filament. In the *sepN*⁻ and Δ *fraD* mutant filaments, we observed a higher rate of cell death on culture level but also in individual filaments. It was striking that often adjacent cells in SJ mutant filaments lysed collectively within a short time frame. These new data are presented in lines 272-295 and new Fig. 7 as well as new Supplementary Fig 8 and Movies S1-S3. The observation that functional SJs are important to ensure the survival of a multicellular bacterium under stress further emphasizes the striking analogy of SJs to metazoan gap junctions.

References

- Battesti A, Bouveret E. 2012. The bacterial two-hybrid system based on adenylate cyclase reconstitution in *Escherichia coli*. *Methods* 58: 325-34
- Merino-Puerto V, Mariscal V, Mullineaux CW, Herrero A, Flores E. 2010. Fra proteins influencing filament integrity, diazotrophy and localization of septal protein SepJ in the heterocyst- forming cyanobacterium *Anabaena* sp. *Molecular Microbiology* 75: 1159-70
- Mourão MA, Hakim JB, Schnell S. 2014. Connecting the dots: the effects of macromolecular crowding on cell physiology. *Biophysical journal* 107: 2761-66
- Oikonomou CM, Jensen GJ. 2017. Cellular Electron Cryotomography: Toward Structural Biology In Situ. *Annu Rev Biochem* 86: 873-96
- Omairi-Nasser A, Mariscal V, Austin JR, Haselkorn R. 2015. Requirement of Fra proteins for communication channels between cells in the filamentous nitrogen-fixing cyanobacterium *Anabaena* sp. PCC 7120. *Proceedings of the National Academy of Sciences* 112: E4458-E64
- Weiss GL, Kieninger A-K, Maldener I, Forchhammer K, Pilhofer M. 2019. Structure and Function of a Bacterial Gap Junction Analog. *Cell* 178: 374-84.e15

Reviewer #1 (Remarks to the Author):

The authors adequately addressed all my critics. I have therefore no other questions to add for the revised manuscript. The new experiments, revealing the role of cell-cell communication in the community survival under stress is spectacular ! This opens a new horizon for the studies on multicellularity In bacteria. Congratulations.

Reviewer #3 (Remarks to the Author):

On one hand, this reviewer maintains that the manuscript falls short of coming up with a definitive model for the molecular architecture of septal junctions. On the other hand, I agree with the authors that it would be very hard, if not impossible to improve resolution allowing integrative model building. As it stands, the work is nevertheless a useful contribution to the field of cell-cell communication in cyanobacteria.